# FoNE: Precise Single-Token Number embeddings via Fourier Features

**Tianyi Zhou**[1]**, Deqing Fu**[1]**, Mahdi Soltanolkotabi**[1]**, Robin Jia**[1]**, Vatsal Sharan**[1]

[1]Department of Computer Science, University of Southern California
{tzhou029,deqingfu,soltanol,robinjia,vsharan}@usc.edu

## Abstract

Language models treat numbers in the same way as ordinary word tokens, which introduces two major issues: (1) embeddings of numerical tokens primarily reflect their frequency in text corpora rather than their inherent numerical properties, leading to frequency bias, and (2) numbers are often split into multiple tokens, forcing the model to aggregate these pieces to recover their values. Inspired by the observation that pre-trained Large Language Models (LLMs) internally learn Fourier-like features for number tokens, we propose **Fo**urier **N**umber **E**mbedding (**FoNE**), a novel method that directly maps numbers into the embedding space with their Fourier features. FoNE encodes each number as a single token with only two embedding dimensions per digit, effectively capturing numerical values without fragmentation. Compared to traditional subword and digit-wise embeddings, FoNE achieves higher accuracy on arithmetic tasks, requires significantly less training data, and offers more efficient training and inference. A 38M-parameter Transformer trained from scratch with FoNE outperforms a fine-tuned Llama-3.2-1B model on addition, subtraction, and multiplication. FoNE is also the only method that achieves 100% accuracy on over 100,000 test examples across these tasks. On 6-digit decimal addition, FoNE needs 64× less data than subword and digit-wise embeddings to reach $\geq 99\%$ accuracy, while using 3× and 6× fewer tokens per number, respectively.

## 1 Introduction

LLMs require precise representations of numerical data to perform number-related tasks effectively. However, since LLMs treat numbers just like any other token, embeddings of numerical tokens do not systematically capture important numerical features. As a result, it is challenging for even billion-parameter models to achieve perfect accuracy in solving simple arithmetic tasks [1] (Saxton et al., 2019; Dziri et al., 2024; Lee et al., 2023; Shen et al., 2023; Zhou et al., 2023a). While generating code can be a useful workaround, relying solely on this capability highlights a fundamental limitation: without a proper understanding of numbers, the model cannot fully grasp concepts critical to domains like mathematical theorems, physics laws, or quantitative reasoning. Even with approaches like Chain-of-Thought (CoT) prompting (Wei et al., 2022), it is important to have a perfect accuracy in solving basic arithmetic tasks to build a strong foundation for more complex reasoning.

Standard tokenization approaches, such as subword tokenization (e.g., GPT-4o Achiam et al., 2023, Llama-3 (Dubey et al., 2024), Phi-2 (Abdin et al., 2024)) or digit-wise tokenization (e.g., Llama-2 (Touvron et al., 2023), Mistral (Jiang et al., 2023)), require the model to aggregate multiple tokens to understand numbers and introduces inefficiencies by tokenizing one number into multiple tokens. However, this inefficiency in tokenizing numbers leads to larger challenges when it comes to their representation. Numbers, unlike words, require systematic, frequency-agnostic representations, yet LLMs often exhibit a frequency bias (Razeghi et al., 2022; Shrestha et al., 2025; Shao et al., 2025), predicting numbers based on training data prevalence rather than their mathematical properties.

---

[1]Our evaluation (See Appendix E.1) of recently released LLMs on arithmetic confirms this limitation: they still struggle with multi-digit addition and multiplication.

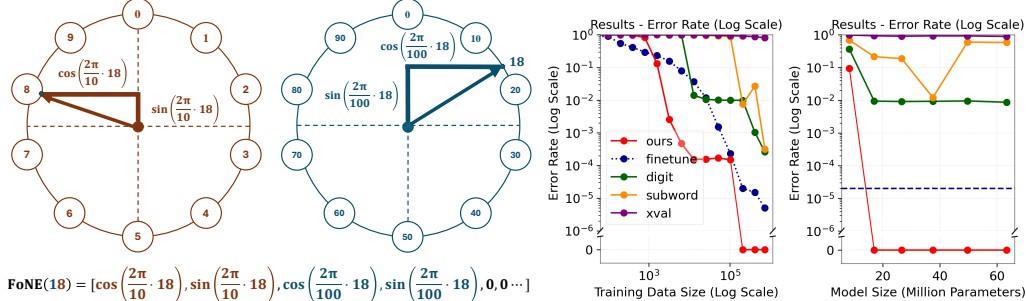

Figure 1: Overview of Fourier Number Embedding (FoNE). Left: FoNE encoder illustrated with the token '18', directly mapped into its FoNE. Middle: Test error on 6-digit decimal addition as the size of the training set increases. Right: Test error on the same task as model size increases. In both plots, we train transformers from scratch with different embedding or tokenization methods until convergence and report the final error. We compares FoNE (ours) against digit-wise tokenization, subword tokenization, XVAL (Golkar et al., 2023), and a fine-tuned Llama-3.2-1B model. FoNE achieves higher accuracy with less data and model size, even surpassing the finetuned Llama baseline

We draw inspiration from interpretability analyses of LLMs, which reveal that models internally develop Fourier-like features. Specifically, pre-trained models embed number tokens using a sparse set of features in the Fourier domain (Zhou et al., 2024). These features enable the representation of numbers capturing both the magnitude and exact values of numbers, which are critical for solving arithmetic tasks (Zhou et al., 2024). However, because numbers are split into subwords and their embeddings are learned from co-occurrence statistics in text during pre-training, current LLMs fail to learn precise numerical representations and struggle to extend these mechanisms to larger numbers, underscoring the need for more systematic approaches to numerical representation.

In this paper, we propose a novel approach called Fourier Number Embedding (FoNE), which directly maps numbers to their Fourier representations, bypassing the tokenization step entirely. By representing each digit using *cosine and sine functions with different periods*, as shown in the left panel of Figure 1, FoNE ensures precise representation of numbers. FoNE represents each digit using only two dimensions in the embedding vector. This compact design not only reduces computational overhead but also creates opportunities for future extensions by incorporating additional features to better capture numeric properties. By embedding and predicting numbers directly as single tokens, our method eliminates the need for multiple forward passes and token aggregation, significantly enhancing computational efficiency. Furthermore, we provide a theoretical justification for why FoNE can represent numbers accurately as single tokens, leveraging the modular encoding properties of trigonometric functions to ensure exact recovery of each digit through periodic embeddings.

Beyond theoretical justification, we demonstrate the effectiveness of FoNE through extensive experiments on arithmetic tasks, including addition, subtraction, and multiplication. Our results show that FoNE is the only approach which—when used to train a Transformer from scratch—achieves perfect accuracy on multiple arithmetic tasks while requiring significantly less training data and fewer model parameters compared to existing methods. Moreover, FoNE offers faster training and inference times by encoding each number into a single token. On 6-digit decimal addition it achieves $\geq 99\%$ accuracy using $64\times$ less data than subword or digit-wise embeddings, while cutting token usage per number by $3\times$ and $6\times$, respectively. These findings underscore FoNE's capacity to represent and manipulate numerical data both efficiently and precisely within large language models.

## 2 RELATED WORK

**Arithmetic and Number-Related Tasks in LLMs.** Using language models for number-related tasks, including solving math problems (Saxton et al., 2019; Yu et al., 2023; Meidani et al., 2023), time-series prediction (Tan et al., 2024; Ma et al., 2024; Zhou et al., 2023b; Liu et al., 2024a; Jin et al., 2023; Cao et al., 2023; Li et al., 2025), quantitative reasoning (McLeish et al., 2024b; Liu et al., 2024b; Chen et al., 2023; Jin et al., 2024; Cobbe et al., 2021), and handling tabular data (Gao et al., 2024; Fang et al., 2024; Sahakyan et al., 2021), remains a significant challenge. Despite advancements in transformer-based models, LLMs such as Qwen3-235B and GPT-5, with billions of parameters, struggle to solve simple arithmetic problems involving multi-digit addition and multiplication across

multiple forward passes (Dziri et al., 2024; Feng et al., 2024), even when using scratchpads (Nye et al., 2021).

Golkar et al. (2023); Sundararaman et al. (2020); Jiang et al. (2019); Sivakumar & Moosavi (2024), introduce number embedding methods to enhance model performance on number-related tasks. However, the range of numbers these methods can accurately represent is typically limited to fewer than five digits and fail to achieve perfect accuracy. Additionally, a line of research (McLeish et al., 2024a; Shen et al., 2023) incorporates the positional information of digits into embeddings or adds it as extra tokens (Nogueira et al., 2021). Lee et al. (2023) demonstrate that smaller transformer models can successfully handle multiplication when equipped with carefully designed scratchpads. However, these approaches are tailored specifically for arithmetic tasks and are difficult to integrate seamlessly into general-purpose LLM training. Thawani et al. (2021) explores encoding strategies like digit-by-digit, scientific notation, and base-10 formats, while Jiang et al. (2019) maps numbers to finite "prototype numerals". These methods help the model align digits of equal significance but often require digit-wise tokenization and introduce additional tokens, reducing training and prediction efficiency. In contrast, the method proposed in this paper precisely encodes all numbers as a single token, eliminating range limitations and avoiding the efficiency drawbacks associated with previous approaches (see Section 5 for further details).

## 3 METHODS

Building on insights from prior studies (Zhou et al., 2024) that highlight the importance of Fourier features in numerical embeddings, we propose Fourier Number Embedding. Unlike existing methods that often require digit-wise tokenization or pre-training to handle numeric tasks, FoNE directly maps numbers into compact Fourier representations. Sections 3.1, 3.3, and 3.4 describe our embedding, decoding, and integration methods, respectively. The complete process is shown in Figure 2.

### 3.1 FOURIER NUMBER EMBEDDING (FONE)

We first introduce the following function that maps each $x \in \mathbb{R}$ to a point on the unit circle.

**Definition 3.1** (Circular embedding). *Let $T$ be a given period. We define the function $\phi : \mathbb{R} \to \mathbb{R}^2$ as*

$$\phi(x, T) := \left( \cos\left( \frac{2\pi}{T} x \right), \sin\left( \frac{2\pi}{T} x \right) \right).$$

Next, we formally define FoNE, which directly maps any floating point number $x$ to an embedding. We predefine $m$ and $n$ as the maximum number of digits before and after the decimal point, respectively. Note that $m$ and $n$ are global hyperparameters fixed for the entire dataset or model, rather than chosen per number.

**Definition 3.2** (Fourier Number Embedding). *Let $m$ be the integer digit length, and $n$ be the decimal digit length. We define the Fourier Number Embedding function $\mathsf{FoNE} : \mathbb{R} \to \mathbb{R}^{2(m+n)}$ for an input number $x$ as follows:*

$$\mathsf{FoNE}(x, m, n) := \left[ \phi(x, T_{-n+1}); \phi(x, T_{-n+2}); \ldots; \phi(x, T_m) \right],$$

*where $T_i = 10^i$ for each integer $i$ in the range $-n + 1$ to $m$.*

To align the embedding dimensions of FoNE with the model's input embedding dimension $d$, we map the Fourier Number Embedding, which lies in $\mathbb{R}^{2(m+n)}$, to $\mathbb{R}^d$. This mapping can be achieved in two ways: (1) by applying a learnable linear transformation $\mathbf{W} \in \mathbb{R}^{d \times 2(m+n)}$, or (2) by appending zeros to the embedding vector to match the dimensionality $d$. As demonstrated in Section 4.3, both approaches achieve comparable results.

### 3.2 FONE'S REPRESENTATIONAL PROPERTIES

Then, we introduce an elementary lemma and demonstrate why FoNE can preserve the numeracy on numbers.

**Lemma 3.3** (Informal version of Lemma D.1). *Given the pair $\left( \cos\left( \frac{2\pi}{T} x \right), \sin\left( \frac{2\pi}{T} x \right) \right)$, we can recover $x \bmod T$.*

Hence, by applying Lemma 3.3 to each frequency component in FoNE, we immediately obtain the following result.

**Lemma 3.4** (FoNE preserves numeracy). *Given a number's Fourier Number Embedding $\mathrm{FoNE}(x)$, its integer digit length $m$, and the decimal digit length $n$, by using Lemma 3.3, we can recover $x \bmod 10^i$ for each integer $i$ in the range $-n + 1$ to $m$.*

A natural question that arises here is the need for $x \bmod 10$, if we already know $x \bmod 100$. The reason is that even though knowing $x \bmod 100$ exactly suffices to recover $x \bmod 10$, this estimation is noisy in practice. When $T$ becomes very large in a circular embedding (Definition 3.1), the difference $\frac{2\pi}{T}(x+1) - \frac{2\pi}{T}x$ approaches zero, causing the embedded representations of $x$ and $x+1$ to become arbitrarily close on the unit circle. Consequently, a single large $T$ cannot sufficiently distinguish adjacent values in the embedding. Hence, one must choose $T$ across a broad range of scales to ensure that the embedding remains adequately distinguishable for all values of $x$. In this paper, we choose $T$ as $10^i$, $\forall i$, so that each $T$ effectively captures one digit of $x$.

To provide a clear illustration of our method, we present a detailed example demonstrating how we map number $4.17$ to its embedding.

**Example 3.5.** *Consider $x = 4.17$. Its Fourier Number Embedding is given by*

$$[\phi(4.17, 0.1); \phi(4.17, 1); \phi(4.17, 10)],$$

*where $\phi$ is defined in Definition 3.1. From these components, by using Lemma 3.3, we can recover*

$$[4.17 \bmod 0.1, 4.17 \bmod 1, 4.17 \bmod 10],\,^2$$

*which simplifies to $[0.07, 0.17, 4.17]$. If we used only $T = 10$, then $\phi(4.17, 10)$ would be nearly indistinguishable from $\phi(4.18, 10)$, causing the embedding to lose fine-grained information about less significant digits. However, with these chosen periods $T$, we can capture all the digits.*

### 3.3 DECODING

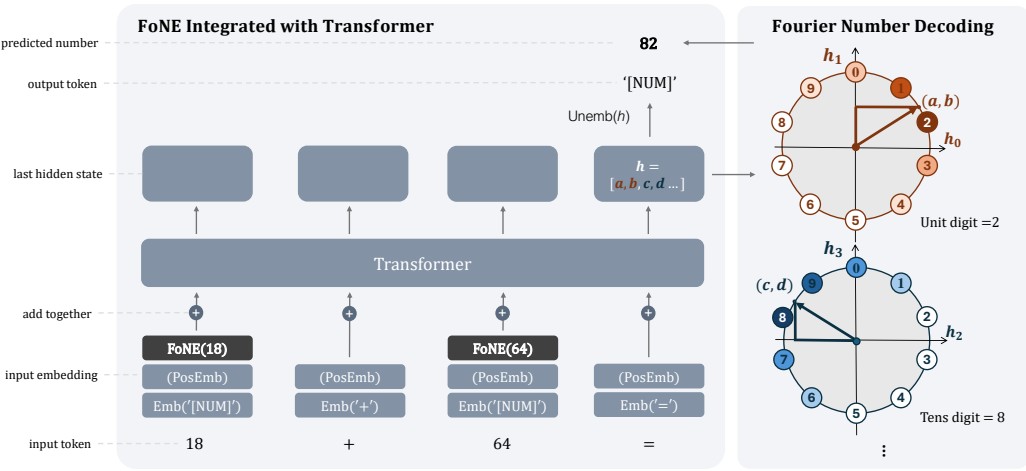

Figure 2: Left: Each number in the input sequence is replaced by a special token [NUM] and embedded as the sum of [NUM] token embedding, its FoNE (see Figure 1(a)), and the standard position embedding (if used by the architecture; Llama-3.2 does not use an explicit position embedding). Right: At decoding time, every pair of hidden-state entries corresponds to one digit, e.g. the first two entries $h_0$ and $h_1$ correspond to the unit digit. The model identifies the digit whose circular Fourier representation best matches those two entries. Digits are then combined by their positional weights.

As each number has its own FoNE, calculating the logits for all possible numbers becomes computationally infeasible. Therefore, we introduce a novel decoding head that maps hidden states from

---

[2] For real $x$ and positive real $m$, $x \bmod m$ is defined as $x - m \cdot \lfloor \frac{x}{m} \rfloor$, yielding a value in the range $[0, m)$

Fourier space to number space as shown in Figure 2. Below, we explicitly define the loss function and prediction function for each digit and then show how to combine these to obtain the final loss and prediction.

**Definition 3.6** (Fourier Number Loss Function). *Let $h \in \mathbb{R}^d$ denote the last-layer hidden state of the model. Let $y_i$ denote the $i$-th digit of the label number $y$. Let $L_{\mathrm{CE}}$ denote the cross entropy loss. For digit $i$, we define the Fourier Number Loss Function $L_{\mathsf{FoNE}}$ as:*

$$L_{\mathsf{FoNE}}(h, y, i) := L_{\mathrm{CE}}\Big(y_i, (\underbrace{[h[2i],\ h[2i+1]]}_{1\times 2} \cdot \underbrace{[\phi(0,10); \cdots ; \phi(9,10)]^\top}_{2\times 10})\Big).$$

Note that the comparison set $\{\phi(0,10), \ldots, \phi(9,10)\}$ is the same for all digit positions $i$. This is because each digit, regardless of its positional significance, must be classified as one of the 10 possible values $\{0, 1, \ldots, 9\}$. This design enables parallel decoding where each digit position is treated as a separate classification task over the same 10-class space. The final training loss is obtained by averaging $L_{\mathsf{FoNE}}(h, y, i)$ over all digit positions $i$.

**Definition 3.7** (Fourier Number Prediction for the $i$-th digit). *Let $h \in \mathbb{R}^d$ denote the last-layer hidden state of the model. For digit $i$, we define the Fourier Number Prediction as:*

$$\widehat{y}_i := \arg\max_{j \in \{0, \ldots, 9\}} \Big( [h[2i],\ h[2i+1]] \cdot [\phi(j, 10)] \Big).$$

Here, $\widehat{y}_i$ is determined by the similarity between the hidden states and the circular embedding of number in $\{0, \cdots, 9\}$ as illustrated in Figure 6(d). Once we have computed $\widehat{y}_i$ for each digit $i$, the final prediction for the entire number can be formed by concatenating these digit-wise predictions. We defer the detailed algorithms to Appendix A.

## 3.4 INCORPORATING FONE INTO INPUT SEQUENCES

To incorporate FoNE, we create one special token [NUM] and add it to the vocabulary. This token must be generated by the model in order to generate any number. We can then remove any tokens corresponding to numbers from the vocabulary. In practice, numbers appear in text with varying formats (e.g., "1,234.56", "$99.99", "3.14e-2"). FoNE handles this diversity by first applying a standard numeric-string parser that detects numbers, ensuring consistent representation regardless of the original textual format.

**Integration Procedure.** The integration of FoNE into input sequences proceeds as follows, as illustrated in Figure 2 and Figure 6:

1. Extract all numbers from the input sequence using the numeric parser to create a number list. Each detected number is replaced with [NUM] and its parsed canonical value. Tokenize the modified sequence to obtain a token list.

2. Embed the token list using standard word embedding methods.

3. Map each canonical number value from the number list to its FoNE representation using Algorithm 1 (Section 3.1).

4. Add the FoNE to the word embedding of the corresponding [NUM] token.

5. Feed the combined embeddings into the model.

6. Use the model's output embeddings to predict the next token in the sequence.

7. If the predicted token is [NUM], decode the numerical value using the method described in Section 3.3, or compute the loss during training.

This procedure ensures that FoNE embeddings are seamlessly integrated into the input sequence. Transformer architectures can effectively aggregate information across tokens; with FoNE, each numeric token carries a precise numerical representation, enabling the model to aggregate and interpret numerical information more reliably than with frequency-biased standard embeddings.

## 4 EMPIRICAL EVALUATION

### 4.1 EXPERIMENTAL SETTING

We evaluate the performance of our proposed *FoNE* method on arithmetic tasks designed to benchmark different number embedding methods. The dataset includes tasks such as 6-digit integer

addition, 6-digit decimal addition (with 3 digits after the decimal), 5-digit integer subtraction, 3-digit integer multiplication, and 4-digit integer multiplication. These tasks are curated to measure model capabilities in accurate numeric computation, while remaining within ranges where baseline embedding methods are still competitive—since their performance degrades rapidly for larger numbers. To further probe the scalability of our approach, we additionally evaluate FoNE on 60-digit addition in Section 4.4, which highlights its ability to handle much larger operands where other embeddings fail.

**Dataset.** Each example in the dataset is formatted as `[operand a][operator][operand b]=`, where the operands `a` and `b` are sampled based on the operation type. For addition and multiplication, we ensure $a \leq b$ to avoid duplication (e.g., $a + b$ and $b + a$ are treated as identical and included only once). For subtraction, we enforce $a \geq b$ to ensure non-negative results. For an $x$-digit operands dataset, each operand can have up to $x$ digits. The dataset is divided into training, validation, and test subsets as shown in Table 10 in Appendix H.

**Baselines.** We compare our proposed FoNE method against several baseline methods for numeric embeddings. First, we consider digit-wise tokenization, where each digit in a number is treated as an individual token. Second, we evaluate subword tokenization, where numeric values are tokenized into subword units based on the Llama3.2-1b tokenizer's default vocabulary. Third, we include the XVAL method Golkar et al. (2023), which leverages explicit value-based representations for numeric computation. As XVAL predict floating point numbers, predictions are rounded to calculate accuracy. Finally, we fine-tune pre-trained LLMs on the same dataset for comparison.

**Setup.** Our experiments involve multiple configurations of randomly initialized transformer-based models. Models were evaluated across varying sizes, ranging from small to large architectures as defined in Table 12. For the accuracy vs. training data size experiments, we use a configuration similar to Llama-3.2 but with 38M parameters. In Appendix I, we conduct more experiments on a transformer model with a different configuration and observe consistent results.

Learning rates were determined through an extensive search, with the best rates selected separately for each method based on the validation performance. Model evaluation used exact match accuracy to assess numeric prediction correctness. All models were trained from random initialization , except the fine-tuned Llama-3.2-1B baseline model. We varied the training data size by uniformly sampling subsets and adjusted model sizes to compare accuracy across methods.

## 4.2 EXPERIMENT RESULTS

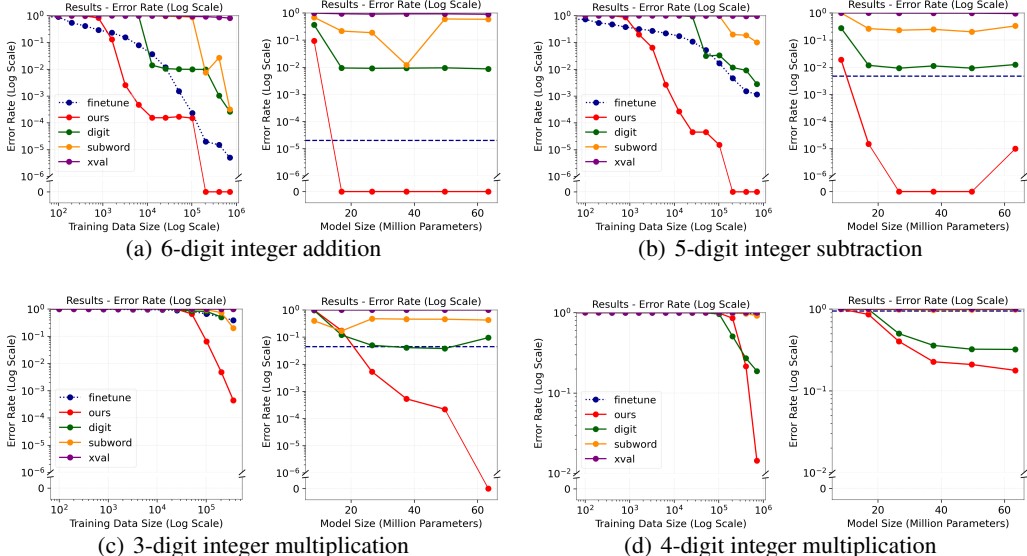

(a) 6-digit integer addition

(b) 5-digit integer subtraction

(c) 3-digit integer multiplication

(d) 4-digit integer multiplication

Figure 3: Comparison of accuracy for various arithmetic tasks with respect to model and data size.

**Data Efficiency.** The middle panel of Figure 1 illustrates the accuracy trends of different embedding methods as the data size increases for the 6-digit decimal addition task. Remarkably, our model achieves 99% accuracy with just $6,400$ training samples and 37.55 million parameters, requiring $64\times$ less training data than traditional embedding methods ($409,600/6,400 = 64$). Even with only $3,200$ training samples, our method outperforms the fine-tuned Llama-3.2 model. Additionally, it achieves perfect accuracy with $51,200$ training samples.

Beyond synthetic tasks, our approach also improves compute efficiency in real-world scenarios. For instance, FoNE requires only 149.25 tokens on average to represent numerical values from a table in the WikiTableQuestions dataset (Pasupat & Liang, 2015), compared to 329.7 tokens used by a digit-wise tokenizer. This significant reduction in token usage highlights the efficiency of our method in encoding numerical data, making it more scalable for number-heavy tasks.

**Parameter Efficiency.** The right panel of Figure 1 shows the accuracy trends of different embedding methods as the model size increases for the 6-digit decimal addition task. Our method achieves 97% accuracy with just 1 layer and 8.31 million parameters using $200k$ examples for training. Furthermore, with 26.62 million parameters, it surpasses the fine-tuned Llama-3.2 model and achieves 100% accuracy.

**Different Tasks.** We conducted the same experiments across all different datasets. As shown in Figure 3, our method consistently demonstrates superior data and parameter efficiency compared to other approaches. Notably, it is the only method that achieves perfect accuracy on 6-digit decimal addition, 6-digit integer addition, 5-digit subtraction, and 3-digit multiplication. We also show that our method performs better in a binary classification task that involves numerical values in Figure 7 and Figure 8 . Specifically, the task requires predicting a label based on a linear equation applied to three integers. In addition, we evaluate FoNE on a modular addition task. In Table 8, we show that it outperforms standard tokenization methods, especially under large moduli where conventional approaches fail. Due to space limitations, we defer the details to Appendix B.

**Training and Inference Efficiency.** Table 1 compares the training and test times used for one epoch across different embedding methods. Our method is consistently faster than digit-wise and subword embedding methods, as it uses one token to embed each number. Compared with XVAL, our method consistently achieves higher accuracy. Additionally, we show the number of tokens required to tokenize the maximum number for different methods, highlighting the efficiency of our approach.

Table 1: Training and inference efficiency comparison across three arithmetic tasks. The times are reported in minutes ($'$) and seconds ($''$).

| Method | Decimal Addition | | | | Subtraction | | | | Multiplication | | | |
| | Train Time | Test. | Tokens | Accuracy | Train. | Test. | Toks. | Acc. | Train. | Test. | Toks. | Acc. |
|---|---|---|---|---|---|---|---|---|---|---|---|---|
| **Ours** | $3'18''$ | $29''$ | 1 | 100 | $2'42''$ | $29''$ | 1 | 100 | $2'56''$ | $33''$ | 1 | 98.56 |
| Digit-wise | $11'48''$ | $1'25''$ | 7 | 99.85 | $9'41''$ | $1'15''$ | 5 | 99.71 | $10'11''$ | $1'18''$ | 8 | 81.21 |
| Subword | $6'46''$ | $58''$ | 3 | 97.94 | $5'47''$ | $54''$ | 2 | 91.66 | $6'20''$ | $58''$ | 3 | 8.05 |
| XVAL | $3'17''$ | $27''$ | 1 | 0.44 | $2'54''$ | $27''$ | 1 | 3.41 | $2'56''$ | $26''$ | 1 | 0 |

## 4.3 ABLATION STUDIES

**Linear Layer after FoNE.** As discussed in Section 3.1, we evaluate the use of a linear layer applied after FoNE and compare it with the approach of appending zeros to align the embedding dimensions with the model's input requirements. As shown in Table 2, both configurations achieve almost the same accuracy. Hence, either technique can be used to align FoNE with the embedding dimension.

Table 2: Accuracy Comparison Across Datasets

| Task | Linear | Zero Padding |
|---|---|---|
| Decimal Addition | 100% | 100% |
| Integer Addition | 100% | 100% |
| Multiplication | 99.95% | 99.91% |
| Subtraction | 100% | 100% |

Table 3: Accuracy Comparison Across Periods

| Dataset | 2,5,10 | 10 | 5 | 7 |
|---|---|---|---|---|
| Decimal Addition | 100 | 100 | 1.52 | 3.64 |
| Integer Addition | 100 | 100 | 1.55 | 0.02 |
| Multiplication | 99.99 | 99.95 | 3.67 | 1.91 |
| Subtraction | 100 | 100 | 4.64 | 0.24 |

**Effect of Different Periods.** As discussed in Section 3.1, the modular group captures the necessary information for each digit, ensuring the effectiveness of our approach. We test the model with base periods of $[2, 5, 10]$, $[5]$, and $[7]$, as shown in Table 3. The $[2, 5, 10]$ configuration achieves accuracy comparable to that of the 10-period setup across different datasets. In this paper, we choose single 10 to make it more parameter efficient. However, configurations using only $\mod 5$ or $\mod 7$ exhibit significantly lower accuracy. This is because neither $\mod 5$ nor $\mod 7$ can fully represent the required information for all digits.

The mispredictions are attributed to the absence of critical modular information. As illustrated in Table 13 in Appendix H, in the decimal addition task, using only a $\mod 5$ representation prevents the model from distinguishing between certain digits, such as 2 and 7, which results in errors.

**Necessity of Sine and Cosine Encoding.** A natural question arises: are sinusoidal encodings truly necessary for arithmetic tasks? One could directly encode each digit into a separate dimension of the embedding, representing a number like 567 as $[5, 6, 7]$. However, this approach fails to achieve perfect accuracy. For instance, numbers such as 999 and 888 become nearly indistinguishable after layer normalization, which reduces their differences and can lead to confusion during training. We evaluate this direct encoding method on 6-digit decimal addition and, after performing a learning rate search, find that the best accuracy is 99.3% with a learning rate of 0.01 and training for 100 epochs. In contrast, FoNE achieves better accuracy in just 6 epochs with the same dataset and model size. This suggests that naive direct encoding does not adequately preserve numerical distinctions for reliable arithmetic operations. As illustrated in Table 14 in the appendix, the model frequently mispredicts 8 as 9, further demonstrating the limitations of direct encoding in preserving numerical structure.

## 4.4 APPLICATIONS AND COMPLEMENTARITY OF FONE

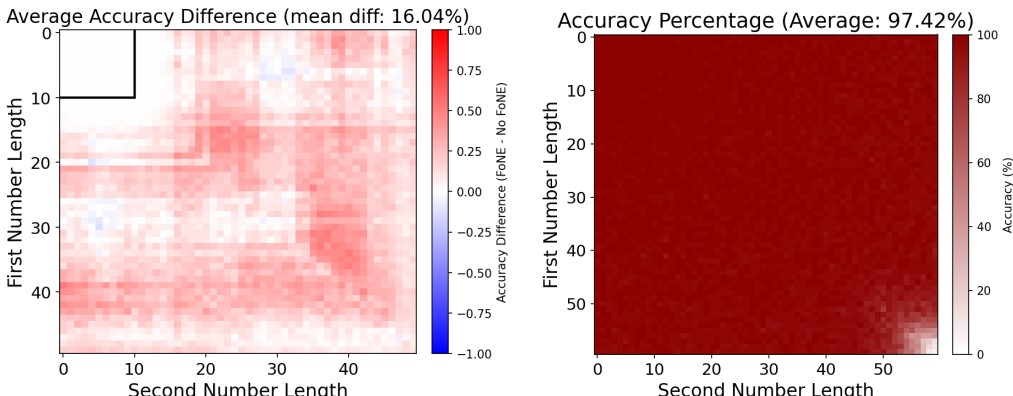

(a) Impact of combining FoNE with Abacus embedding    (b) Test accuracy of 60-digit addition with FoNE

Figure 4: (a) Performance improvements achieved by combining FoNE with the Abacus embedding method across various random seeds. The transformer is trained on addition tasks with up to 10-digits numbers (represented by the smaller square) and tested up to 50-digit numbers. (b) Average accuracy of an 8-layer transformer model on 60-digit addition tasks using FoNE for chunked input.

**Combining FoNE with Abacus.** FoNE can be combined with other positional embedding methods even with different tokenization methods, requiring only minor modifications. For instance, we integrated FoNE with the Abacus embedding method (McLeish et al., 2024a), which operates on digit-wise tokenization. In this setup, the embeddings for each digit (0–9) are replaced with their corresponding FoNE. We trained an 8-layer transformer model on integer addition tasks with up to 10 digits and tested it on addition tasks involving up to 50-digit numbers. The results, as illustrated in Figure 4(a) and Figure 12 in Appendix G, show that incorporating FoNE consistently improves the performance of the Abacus method across various random seeds. This highlights the complementary benefits of combining FoNE with other positional embedding strategies.

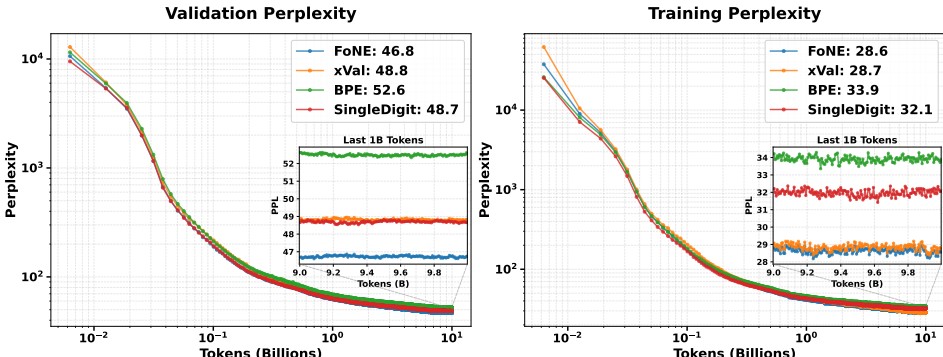

Figure 5: Pretraining 117M GPT-2 model on 10 Billion tokens with various number tokenization methods. Our method (FoNE) achieves the lowest validation perplexity. This experiment shows that FoNE does not harm the language models' semantic abilities.

**How does FoNE Handle numbers with longer digit sequences** The maximum digit length that a `float64` data type can represent is 15 digits. When $x$ exceeds 15 digits in length, applying $\text{FoNE}(x)$ directly may result in a loss of precision. To address this, $x$ can be divided into smaller chunks, and FoNE can be applied to each chunk independently. For example, $x$ can be split into groups of five digits. The FoNE can then be calculated for each chunk, resulting in a representation of length 10 per chunk, as each digit is encoded in two dimensions. These embeddings are subsequently concatenated to obtain the final number embedding for $x$. Note that we are still using one token for each number. By using this method, as shown in Figure 4(b), an 8-layer transformer trained on 60-digit addition achieved an average accuracy of 97.42% across different operand length with just one forward pass. This demonstrates the effectiveness of FoNE in handling long sequences.

## 5 DISCUSSION

*Q1:* **Why not use regression loss instead of classification loss, which minimizes RMSE and can yield smaller prediction errors (Zausinger et al., 2024).** The key limitation is that regression produces continuous values, making it impossible to integrate number-related tasks with general language modeling. For example, when predicting the year "1997", regression may output "1996.9999", which is acceptable under regression metrics but unusable in sequence generation or token-based reasoning. In contrast, FoNE retains a classification-based loss, so it does not require explicitly identifying which numbers are used for arithmetic. This ensures seamless compatibility with standard LLM training while still delivering accurate numerical representations.

*Q2:* **Why does FoNE outperform other number embedding methods?** Note that since FoNE uses the ratio between entries to represent numbers as shown in Lemma 3.4, it is unaffected by layer normalization and RMS normalization (Lemma D.2), in contrast to xVal (Golkar et al., 2023), which uses the magnitudes of entries. Other approaches, such as DICE (Sundararaman et al., 2020) and SALSA (Stevens et al., 2024), map numbers onto a single unit circle, which can limit their capacity to distinguish between different magnitudes effectively. FoNE, by leveraging multiple sinusoidal components, captures both the magnitude and periodicity of numbers more precisely. This comprehensive representation enables FoNE to achieve higher accuracy and generalization in number-related tasks.

*Q3:* **Why do we choose components with periods that are multiples of** $10$**, i.e.,** $10, 100, 1000 \cdots$**?** As shown in Zhou et al. (2024) and Figure 10, pre-trained LLMs trained on diverse datasets and strategies consistently learn nearly identical key frequency components. These components have periods of 10 and its divisors, such as 2 and 5. Since $\mod 10$ can already represent a single digit, we believe that $\mod 2$ and $\mod 5$ contribute to enhancing robustness. Models trained on real-world text data—where numbers are almost always expressed in decimal—commonly learn frequency components that correspond to $\mod 10$. In principle, we could choose alternative bases (such as 5, 16, etc.) to help the model better learn arithmetic in those bases, as demonstrated in Table 7. However, since most large language models primarily encounter numbers in base 10, and our results show that

base-10 FoNE already performs well on arithmetic tasks in other bases (Table 6), we adopt base 10 as the default. Additional experiments validating this choice are provided in Appendix B.

*Q4:* **Can FoNE be integrated into pretrained LLMs without harming their semantic abilities?**

We study this question in two complementary settings. First, we perform continual pretraining of an existing LLM with a simplified FoNE adaptation. Second, we train a smaller GPT-2–117M model from scratch on 10B FineWeb tokens with different number tokenization schemes. In both cases, FoNE preserve, and in the latter case even improving, language modeling performance.

**Continual pretraining on Llama-3.1-1B.** We demonstrate that our FoNE token embedding can be merged with any existing LLM with slight continual pretraining. We made a simplification of FoNE where instead of overriding all the number embeddings which could make continual pretraining harder, we build the simplified FoNE on top of existing BPE tokenization. We compute FoNE embedding of each subword, and continual pretrain a linear projection layer from the FoNE embedding space to the original token embedding space to align the embeddings similar to vision language models' alignment phase. We continual pretrain both the original Llama-3.1-1B model and our FoNE adapted version on 15B tokens from `MegaMath-Web-Pro` (Zhou et al., 2025) and evaluated on arithmetic tasks from Brown et al. (2020) and on MMLU (Hendrycks et al., 2021). As shown in Table 4, this simplified version of FoNE improves the model's zero-shot arithmetic with larger number of digits without affecting language abilities evaluated by MMLU.

Table 4: A simplified version of FoNE improves model's zero-shot arithmetic abilities without sacrificing language abilities.

| TASK | REGULAR | FONE (*Simplified*) |
|---|---|---|
| 4-digit addition | 51.35% | **59.00**% |
| 4-digit subtraction | 29.60% | **39.90**% |
| 5-digit addition | 29.40% | **35.75**% |
| 5-digit subtraction | 24.95% | **33.95**% |
| MMLU | 38.10% | 38.21% |

Table 5: Perplexity of text tokens only on sequences that contain both text and numbers. (evaluated on ∼5 million tokens in total).

| Method | Perplexity |
|---|---|
| BPE | 67.390 |
| Single Digit | 55.922 |
| xVal | 49.160 |
| **FoNE** | **46.856** |

**Pretraining from scratch on FineWeb.** To make the model learn text and numbers together, we also pretrain a GPT-2–117M model from scratch on 10 billion tokens from FineWeb (Penedo et al., 2024) under four number tokenization schemes: BPE, Single Digit, xVal, and FoNE. As shown in Figure 5, the FoNE model achieves the lowest validation perplexity on the FineWeb validation set. Furthermore, we evaluate the perplexity of text tokens only on sequences that contain both text and numbers. As shown in Table 5, FoNE remains the lowest perplexity while other methods show even worse perplexity compared to their performance on all sequences. Since numbers in FineWeb appear naturally interleaved with text, this experiment shows that using FoNE throughout pretraining not only preserves the model's semantic abilities but can in fact *improve* its language modeling quality by giving better number representation.

**Semantic vs. numerical roles.** These findings are consistent with prior interpretability results. As discussed by Meng et al. (2022), semantic and factual associations of tokens—such as the historical or cultural meaning of a year—are often stored in the MLP layers of transformer models rather than in the token embeddings themselves. FoNE therefore plays a complementary role: it provides precise, numeracy-preserving input embeddings for numbers, while the model's MLP layers continue to store and retrieve semantic knowledge. Together, this separation of roles allows FoNE to enhance numerical reasoning without sacrificing semantic competence.

## 6 CONCLUSION

In this paper, we introduced FoNE, a novel method for representing numbers in the embedding space of LLMs. By leveraging Fourier features, FoNE directly maps numbers into a compact and precise representation, bypassing tokenization inefficiencies and preserving essential numerical properties. FoNE has significant implications for pre-training LLMs. By incorporating FoNE, models can develop a robust understanding of numerical concepts, addressing a fundamental limitation in current architectures. We believe our work establishes a solid foundation for future research on a wide range of number-related tasks, including time-series analysis, quantitative reasoning, and complex operations in fields like physics and mathematics.

## ACKNOWLEDGMENTS

We are grateful to Wang Zhu for providing helpful feedback on our paper. This work was supported by AWS credits from a gift from the USC-Amazon Center on Secure and Trusted Machine Learning. DF, MS, and RJ were supported in part by a gift from the USC-Capital One Center for Responsible AI and Decision Making in Finance (CREDIF). RJ was also supported in part by the National Science Foundation under Grant No. IIS-2403436. The work of MS was also supported in part by AWS credits through an Amazon Faculty Research Award, a NAIRR Pilot Award, and generous funding by Coefficient Giving. MS is also supported by the Packard Fellowship in Science and Engineering, a Sloan Research Fellowship in Mathematics, NSF CAREER Award #1846369, DARPA FastNICS program, NSF CIF Awards #1813877 and #2008443, and NIH Award DP2LM014564-01. Any opinions, findings, and conclusions or recommendations expressed in this material are those of the author(s) and do not necessarily reflect the views of the National Science Foundation.

## REPRODUCIBILITY STATEMENT

We have taken several steps to ensure the reproducibility of our results. The FoNE method is defined step by step in Section 3.1 and Section A. Our experimental setup, including datasets, sampling rules, model configurations, and training details, is described in Section 4.1 and Appendix H.

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

APPENDIX

# A DETAILED ALGORITHMS FOR COMPUTING FoNE, AND MAKING PREDICTIONS

In this section, we provide the detail pipeline and algorithms of how we compute FoNE, get the final loss and final prediction as defined in Section 3.1.

We first show the how are FoNE and Fourier number decoding integrated with regular Transformer pipeline.

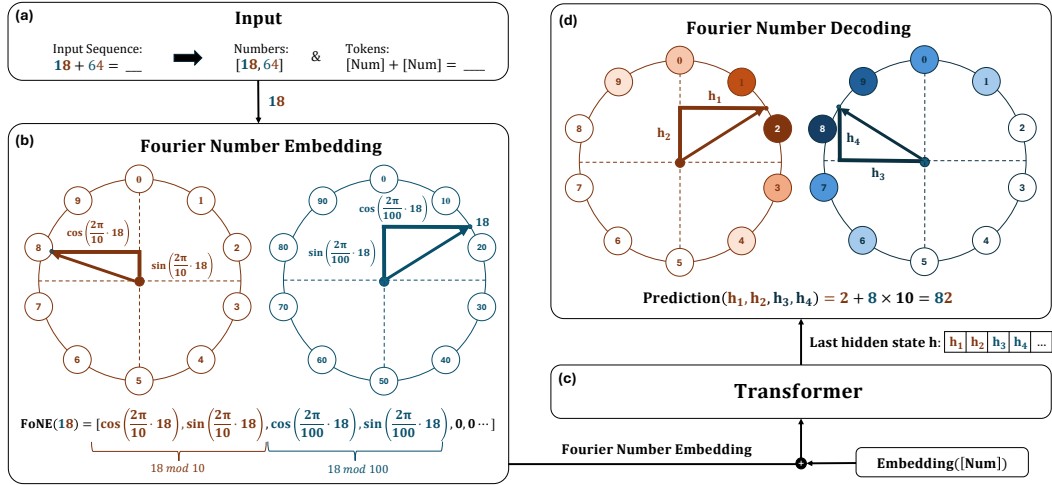

Figure 6: (a) We extract all the numbers from the input sequence. (b) For each number, we use FoNE to directly map the number to its embedding. The first two entries in the embedding represent $18 \bmod 10$, while the next two entries represent $18 \bmod 100$. (c) We pad the FoNE with zeros, add it to the word embeddings, and then feed the combined embeddings into the model. (d) For each digit, we take every two entries from the last hidden state and find the number whose representation is closest to these two entries.

Next we show the exact algorithms we use to compute FoNE, compute loss and make the final prediction.

---

**Algorithm 1** Fourier Number Embedding (FoNE) Algorithm

---

1: **procedure** FOURIERNUMBEREMBEDDING($x \in \mathbb{R}, m \in \mathbb{Z}_{\geq 0}, n \in \mathbb{Z}_{\geq 0}, d \in \mathbb{Z}_{>0}$)
2:    **Inputs**: Number $x$, integer digit length $m$, decimal digit length $n$, embedding dimension $d$
3:    Initialize empty embedding vector FoNE $\leftarrow []$
4:    **for** $i = -n + 1 \rightarrow m$ **do**                          ▷ Loop over all scales from $10^{-n+1}$ to $10^m$
5:        $T_i \leftarrow 10^i$                                          ▷ Set the period for the current scale
6:        $\phi(x, T_i) \leftarrow (\cos(\frac{2\pi}{T_i}x), \sin(\frac{2\pi}{T_i}x))$       ▷ Compute the circular embedding for scale $T_i$
7:        Append $\phi(x, T_i)$ to FoNE                              ▷ Add the embedding for this scale to the result
8:    **end for**
9:    **while** Length(FoNE) $< d$ **do**                        ▷ Ensure embedding dimension matches the target
10:       Append $0$ to FoNE                                                            ▷ Zero-pad
11:   **end while**
12:   **return** FoNE
13: **end procedure**

---

---

**Algorithm 2** Fourier Number Loss & Prediction

---

1: **function** FOURIERNUMBERLOSSFUNCTION($h, y, i$)
2:     $y_i \leftarrow$ the $i$-th digit of $y$
3:     $a \leftarrow \big[h[2i], h[2i+1]\big]$
4:     $b \leftarrow [\phi(0, 10), \phi(1, 10), \cdots, \phi(9, 10)]^\top$
5:     logits $\leftarrow a \cdot b$
6:     loss $\leftarrow L_{\mathrm{CE}}(y_i, \text{logits})$                    ▷ *Cross-entropy loss for digit $i$*
7:     **return** loss
8: **end function**
9: **function** FOURIERNUMBERPREDICTION($h, i$)                    ▷ *Prediction for digit $i$*
10:     logits $\leftarrow \big[h[2i], h[2i+1]\big] \cdot \big[\phi(j, 10)\big]_{j=0,\dots,9}$
11:     $\widehat{y}_i \leftarrow \arg\max_{j \in \{0,\dots,9\}} \text{logits}[j]$
12:     **return** $\widehat{y}_i$
13: **end function**

---

**Algorithm 3** Fourier Number Final Loss & Prediction

---

1: **function** FOURIERNUMBERFINALLOSS($h, y, m, n$)                    ▷ *Compute average loss*
2:     totalLoss $\leftarrow 0$
3:     $\mathcal{I} \leftarrow [m+n]$
4:     **for** $i \in \mathcal{I}$ **do**
5:         digitLoss $\leftarrow$ FOURIERNUMBERLOSSFUNCTION($h, y, i$)
6:         totalLoss $\leftarrow$ totalLoss $+$ digitLoss
7:     **end for**
8:     finalLoss $\leftarrow \frac{\text{totalLoss}}{|\mathcal{I}|}$                    ▷ Average over all digit positions
9:     **return** finalLoss
10: **end function**
11: **function** FOURIERNUMBERFINALPREDICTION($h, m, n$)                    ▷ *Compute final prediction*
12:     $\widehat{y} \leftarrow 0$
13:     $\mathcal{I}_{\mathrm{frac}} \leftarrow [0, \dots, n-1]$                    ▷ Fractional digit indices
14:     $\mathcal{I}_{\mathrm{int}} \leftarrow [n, \dots, m+n-1]$                    ▷ Integer digit indices
15:     **for** $i \in \mathcal{I}_{\mathrm{frac}}$ **do**
16:         $\text{logits}_i \leftarrow \big[h[2i], h[2i+1]\big] \cdot \big[\phi(j, 10)\big]_{j=0,\dots,9}$
17:         $\widehat{y}_i \leftarrow \arg\max_{j \in \{0,\dots,9\}} \text{logits}_i[j]$
18:         $\widehat{y} \leftarrow \widehat{y} + \widehat{y}_i \cdot 10^{-(n-i)}$                    ▷ Scale fractional part by $10^{-(n-i)}$
19:     **end for**
20:     **for** $j \in \mathcal{I}_{\mathrm{int}}$ **do**
21:         $\text{logits}_j \leftarrow \big[h[2j], h[2j+1]\big] \cdot \big[\phi(j, 10)\big]_{j=0,\dots,9}$
22:         $\widehat{y}_j \leftarrow \arg\max_{j \in \{0,\dots,9\}} \text{logits}_j[j]$
23:         $\widehat{y} \leftarrow \widehat{y} + \widehat{y}_j \cdot 10^{j-n}$                    ▷ Scale integer part by $10^j$
24:     **end for**
25:     **return** $\widehat{y}$
26: **end function**

---

## B    FoNE on Binary Classification and Modular Arithmetic Task

In this section, we demonstrate that FoNE outperforms other methods on binary classification tasks and modular arithmetic tasks, benefiting from its precise representation.

**Binary Classification Task**    Each example in the dataset is formatted as [num1, num2, num3], where the integers num1, num2, and num3 are sorted in ascending order ($num1 \leq num2 \leq num3$) to ensure uniqueness and eliminate duplicate representations of the same combination. The integers are uniformly sampled from the range $[0, 1000]$. The label for each example is determined by evaluating the linear equation

$$a \cdot \texttt{num1} + b \cdot \texttt{num2} + c \cdot \texttt{num3} - d,$$

using predefined coefficients $a = 1.5$, $b = -2$, $c = 0.5$, and $d = 10$ and $a = 1.5$, $b = -2$, $c = 0.5$, and $d = -190$. If the result is greater than zero, the label is assigned as $1$; otherwise, it is assigned as $0$. The dataset is divided into training, validation, and test subsets, as outlined in Table 10. Figure 7 and Figure 8 show that FoNE outperforms the regular embedding method, XVAL, and even a fine-tuned Llama-3.2-1B model by requiring less data and achieving higher accuracy.

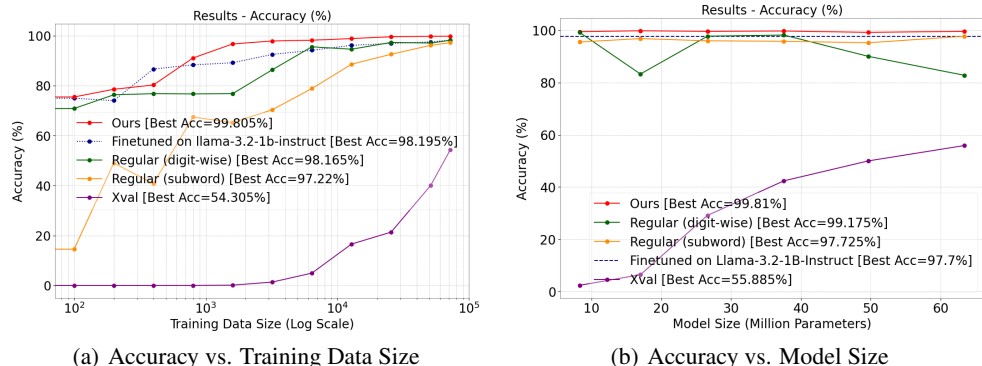

(a) Accuracy vs. Training Data Size      (b) Accuracy vs. Model Size

Figure 7: We train Llama-3.2-1B from scratch with random initialization using different number embedding methods on number classification where $d = 10$. The test accuracy is compared across varying data sizes and model sizes.

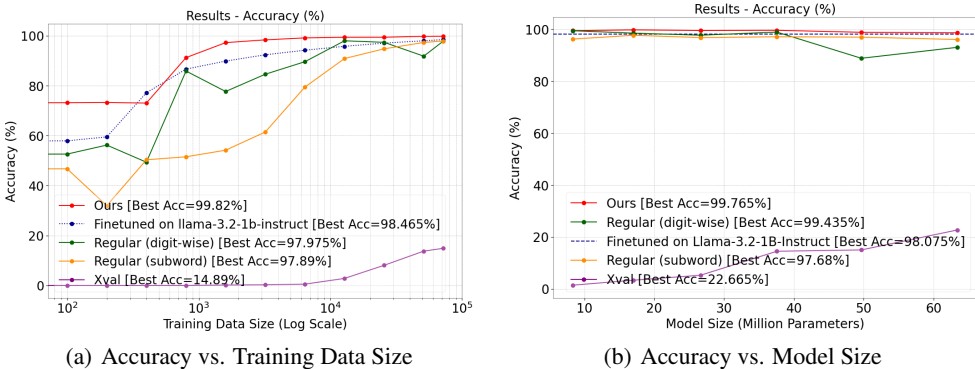

(a) Accuracy vs. Training Data Size      (b) Accuracy vs. Model Size

Figure 8: We train Llama-3.2-1B from scratch with random initialization using different number embedding methods on number classification where $d = -190$. The test accuracy is compared across varying data sizes and model sizes.

**Modular Arithmetic and Base Selection** We conduct experiments varying the FoNE base from 7 to 13 and observed a clear "sweet spot" at base 10. For example, at base 7 the addition, subtraction, and multiplication accuracies are only 17.66 %, 1.93 %, and 0.54 %, respectively, while at base 10 we achieve 100 % accuracy on addition and subtraction after just 4 and 26 epochs, and 99.25 % on multiplication after 50 epochs. Accuracy then fall off again at larger bases (e.g., at base 13: 1.06 %, 5.71 %, and 1.91 %). This confirms that bases that are too small or too large degrade digit-level distinguishability.

We also aligned FoNE's moduli to the numbers' representation base by preprocessing inputs into base 5. In that setting, only the FoNE embedding with period 5 reached perfect accuracy (100 % in 6 epochs), whereas other periods (2–4 and 7–8) yielded near-zero to sub-20 % accuracy (e.g., period 2: 0.25 %; period 3: 0.12 %; period 7: 16.38 %). This again demonstrates that FoNE performs best when its moduli match the underlying digit base.

Table 6: Accuracy (%) on arithmetic tasks under different bases.

| Base | Epochs (Add/Sub/Mul) | Accuracy (Add/Sub/Mul) |
|------|----------------------|------------------------|
| 7 | 50 / 50 / 50 | 17.66 / 1.93 / 0.54 |
| 8 | 50 / 50 / 50 | 40.91 / 6.07 / 0.33 |
| 9 | 50 / 50 / 50 | 65.49 / 65.63 / 65.78 |
| 10 | 4 / 26 / 50 | 100.00 / 100.00 / 99.25 |
| 11 | 50 / 50 / 50 | 99.99 / 99.99 / 78.76 |
| 12 | 50 / 50 / 50 | 99.78 / 99.99 / 54.14 |
| 13 | 50 / 50 / 50 | 1.06 / 5.71 / 1.91 |

Table 7: Accuracy (%) with base-5 number inputs, under varying FoNE periods.

| FoNE Period | Epochs | Accuracy (%) |
|-------------|--------|--------------|
| 2 | 50 | 0.25 |
| 3 | 50 | 0.12 |
| 4 | 50 | 0.17 |
| 5 | 6 | 100.00 |
| 7 | 50 | 16.38 |
| 8 | 50 | 99.97 |

We additionally benchmarked FoNE on modular addition tasks with varying moduli. The goal is to predict $x + y \bmod m$, where $x, y \in [0, m)$. FoNE consistently outperforms standard tokenization, especially at higher moduli.

Table 8: Accuracy (%) on modular addition tasks across various moduli.

| Modulus $m$ | FoNE Accuracy | Standard Tokenization Accuracy |
|-------------|---------------|--------------------------------|
| 11 | 100.00 | 99.98 |
| 60 | 99.27 | 99.99 |
| 97 | 100.00 | 0.89 |
| 100 | 100.00 | 18.17 |
| 113 | 100.00 | 99.99 |
| 121 | 100.00 | 5.48 |
| 225 | 100.00 | 65.10 |
| 256 | 100.00 | 70.20 |
| 257 | 100.00 | 11.36 |

These experiments validate our claim that FoNE is most effective when its moduli align with the target numeral base, and it remains robust even in high-modulus arithmetic settings where standard tokenization breaks down.

## C  MORE RELATED WORK

**Fourier Features.**    Fourier features are commonly observed in image models, particularly in the early layers of vision models Olshausen & Field (1997); Olah et al. (2020); Fiquet & Simoncelli (2024). These features enable the model to detect edges, textures, and other spatial patterns effectively. However, Transformers struggle to capture high-frequency components (Bai et al., 2022; Tancik et al., 2020). Augmenting data with high-frequency components or explicitly encoding coordinates using Fourier features has been demonstrated to improve model performance (Tancik et al., 2020; He et al., 2024; Hua et al., 2024). In fact, the original Transformers paper (Vaswani et al., 2017) uses Fourier features to encode the position information of tokens; however, it does not apply this idea to number tokens to aid with numerical tasks. In modular addition tasks, studies have revealed that after "grokking," a one-layer Transformer can learn to solve the task perfectly by leveraging Fourier features (Nanda et al., 2023; Gu et al., 2024). Furthermore, Zhou et al. (2024) demonstrate that LLMs naturally encode numbers using Fourier features during pretraining, leveraging these representations for arithmetic tasks (Levy & Geva, 2024; Kantamneni & Tegmark, 2025; Lindsey

et al., 2025). Building on this insight, we propose using modular Fourier components to explicitly represent digits, enabling models to perform precise numerical computation. This allows algebraic operations to be carried out in a component-wise, parallel manner and overcomes the limitations of token-based number representations.

# D PRELIMINARIES AND MISSING PROOFS

## D.1 PRELIMINARIES

In this section, we provide the necessary mathematical definitions and concepts used throughout the paper.

**Period and Frequency.** A function $f(x)$ is periodic with period $T > 0$ if $f(x + T) = f(x)$ for all $x$. The period $T$ represents the smallest positive value for which the function repeats. The frequency $f$ of a periodic function is the reciprocal of its period, $f = \frac{1}{T}$, and describes the number of cycles completed in one unit interval. For the sine and cosine functions $\cos\left(\frac{2\pi}{T}x\right)$ and $\sin\left(\frac{2\pi}{T}x\right)$, the period is $T$.

**Unit Circle.** The unit circle is the set of points in the plane at a distance of 1 from the origin, given by $x^2 + y^2 = 1$. The coordinates of points on the unit circle can be parameterized as $(\cos\theta, \sin\theta)$, where $\theta$ is the angle measured counterclockwise from the positive $x$-axis. For any angle $\theta$, $\cos\theta$ represents the $x$-coordinate, and $\sin\theta$ represents the $y$-coordinate.

**Two-Argument Inverse Tangent.** The two-argument inverse tangent function, $\mathrm{atan2}(y, x)$, determines the angle $\theta$ (modulo $2\pi$) given the coordinates $(x, y) = (\cos\theta, \sin\theta)$. Specifically,

$$\theta = \mathrm{atan2}(y, x),$$

which resolves the angle $\theta$ uniquely based on the signs of $x$ and $y$.

**Modular Arithmetic.** Modular arithmetic considers equivalence classes of numbers under a modulus $T > 0$. For integers $a$ and $b$, $a \equiv b \pmod{T}$ if $T \mid (a - b)$, meaning $a$ and $b$ differ by an integer multiple of $T$.

**Fourier Representation.** Periodic functions with period $T$ can be represented using the fundamental frequencies $\frac{2\pi}{T}$. For example, the embeddings $\left(\cos\left(\frac{2\pi}{T}x\right), \sin\left(\frac{2\pi}{T}x\right)\right)$ capture the periodicity of $x$ modulo $T$ by mapping it to a unique point on the unit circle.

## D.2 MISSING PROOFS

In this section, we provide some missing proofs.

**Lemma D.1** (Formal version of Lemma 3.3). *Given the pair $\left(\cos(\frac{2\pi}{T}x), \sin(\frac{2\pi}{T}x)\right)$, we can recover* $x \bmod T$.

*Proof.* Let $\theta = \frac{2\pi}{T}x$. Then the given pair becomes

$$\left(\cos(\theta), \sin(\theta)\right).$$

From this pair, one can recover $\theta$ uniquely modulo $2\pi$. Concretely, $\theta$ can be obtained (modulo $2\pi$) using the two-argument inverse tangent function:

$$\theta \equiv \mathrm{atan2}\left(\sin(\theta), \cos(\theta)\right) \pmod{2\pi}.$$

Since $\theta = \frac{2\pi}{T}x$, we have

$$x = \frac{T}{2\pi}\theta.$$

Hence $x$ is determined up to integer multiples of $T$, i.e., $x \bmod T$.

In other words, if

$$\left(\cos(\tfrac{2\pi}{T}x_1), \sin(\tfrac{2\pi}{T}x_1)\right) = \left(\cos(\tfrac{2\pi}{T}x_2), \sin(\tfrac{2\pi}{T}x_2)\right),$$

then $\frac{2\pi}{T}x_1 \equiv \frac{2\pi}{T}x_2 \pmod{2\pi}$, which implies $x_1 \equiv x_2 \pmod{T}$. Therefore, from the pair $\left(\cos(\frac{2\pi}{T}x), \sin(\frac{2\pi}{T}x)\right)$, we can indeed recover $x \bmod T$. $\qquad \square$

**Lemma D.2** (Layer-Normalized FoNE Preserves Numeracy). *Given a number's Layer-Normalized Fourier Number Embedding* $\text{LN}(\text{FoNE}(x) + \mathbf{p})$, *where* $\text{FoNE}(x)$ *is the Fourier Number Embedding of* $x$ *and* $\mathbf{p}$ *is an orthogonal positional encoding vector, assume the mean of* $\text{FoNE}(x) + \mathbf{p}$ *is 0. Let* $m$ *be the integer digit length of* $x$ *and* $n$ *be the decimal digit length of* $x$. *Then, using Lemma 3.3, we can recover* $x \bmod 10^i$ *for each integer* $i$ *in the range* $-n+1$ *to* $m$.

*Proof.* Assume the mean of $\mathbf{x} = \text{FoNE}(x) + \mathbf{p}$ is 0, i.e., $\mu = 0$. Under this assumption, LayerNorm simplifies to:

$$\text{LN}(\mathbf{x}) = \frac{\mathbf{x}}{\sigma},$$

where $\sigma$ is the standard deviation of $\mathbf{x}$.

Let $\mathbf{u} = \text{FoNE}(x)$ encode the scalar $x$, and let $\mathbf{p}$ be an orthogonal positional encoding vector such that:

$$\|\mathbf{u}\| = \|\mathbf{p}\| = 1 \quad \text{and} \quad \mathbf{u} \cdot \mathbf{p} = 0.$$

Then, the input to LayerNorm is:

$$\mathbf{x} = \mathbf{u} + \mathbf{p}.$$

The standard deviation $\sigma$ of $\mathbf{x}$ is given by:

$$\sigma = \sqrt{\frac{1}{d} \sum_{i=1}^{d} (\mathbf{x}_i - \mu)^2},$$

where $d$ is the dimensionality of $\mathbf{x}$. Since $\mu = 0$, this simplifies to:

$$\sigma = \sqrt{\frac{1}{d} \|\mathbf{x}\|^2}.$$

Substitute $\mathbf{x} = \mathbf{u} + \mathbf{p}$:

$$\|\mathbf{x}\|^2 = \|\mathbf{u} + \mathbf{p}\|^2 = \|\mathbf{u}\|^2 + \|\mathbf{p}\|^2 + 2\mathbf{u} \cdot \mathbf{p}.$$

By orthogonality and unit norm, $\mathbf{u} \cdot \mathbf{p} = 0$, $\|\mathbf{u}\|^2 = 1$, and $\|\mathbf{p}\|^2 = 1$. Thus:

$$\|\mathbf{x}\|^2 = 1 + 1 + 0 = 2.$$

Therefore:

$$\sigma = \sqrt{\frac{1}{d} \cdot 2} = \sqrt{\frac{2}{d}}.$$

The LayerNorm operation simplifies to:

$$\text{LN}(\mathbf{x}) = \frac{\mathbf{x}}{\sigma} = \frac{\mathbf{u} + \mathbf{p}}{\sqrt{\frac{2}{d}}} = \sqrt{\frac{d}{2}}(\mathbf{u} + \mathbf{p}).$$

This rescales $\mathbf{u}$ and $\mathbf{p}$ by a factor of $\sqrt{\frac{d}{2}}$.

The key observation is that LayerNorm applies a **uniform scaling** to all components of $\mathbf{x}$. Since $\mathbf{u}$ and $\mathbf{p}$ are orthogonal and their relative directions are preserved, the numerical relationships encoded in $\mathbf{u}$ (which represent $x$) are preserved up to a scaling factor.

By Lemma 3.3, the numeracy of $x$ is preserved. This means we can recover $x \bmod 10^i$ for all $i$ in the range $-n+1 \leq i \leq m$, as the normalized embedding retains the necessary information about $x$.

$\qquad \square$

The same result holds for RMSNorm because it also applies a uniform scaling (based on the root mean square of the input) while preserving the relative directions of the embedding components, thus maintaining the numeracy of $x$.

Notet that standard sinusoidal positional encodings (PEs) (Vaswani et al., 2017) cannot serve as a substitute for numerical embeddings. PEs are explicitly constructed to distinguish token positions in a sequence, not to represent the magnitude or digit structure of real numbers. They use a fixed set of exponentially spaced frequencies to ensure each position has a unique yet non-invertible signature; as a result, there is no straightforward way to recover the original numeric value (or its individual digits) from a PE embedding. Moreover, the frequencies in PEs are chosen for positional uniqueness, not for digit-aligned modular arithmetic, so small changes in numeric value can produce large, non-monotonic changes in the embedding space—precisely the opposite of the smooth, digit-wise variation required for accurate number encoding. Unlike FNE's digit-aligned sinusoidal components (e.g., mod 10, mod 100, ...), PEs do not guarantee invertibility with respect to numeric operations.

# E  MORE EVIDENCE

## E.1  LLMS STRUGGLE WITH MULTI-DIGIT ARITHMETIC

We evaluate five production LLMs (Claude 3.7 Sonnet, DeepSeek V3.1, Gemini 2.5 Flash, GPT-5, and Qwen3-235B) on direct arithmetic without chain-of-thought, code execution, or tools. For each setting, we sample 100 IID problem instances with uniformly distributed operands over the full $d$-digit numbers. Tasks include $d \in \{3, 4, 5, 6\}$ for multiplication and $d \in \{7, 8, 50\}$ for addition. Models receive a two-shot, operation-matched prompt and must return a single integer only. Decoding uses temperature 0. We parse the first integer token from the response and score exact-match accuracy by numeric equality with the reference.

| Model | Multiplication | | | | Addition | | |
|---|---|---|---|---|---|---|---|
| | 3 | 4 | 5 | 6 | 7 | 8 | 50 |
| Claude 3.7 Sonnet | 1.00 | 0.53 | 0.02 | 0.00 | 1.00 | 0.97 | 0.82 |
| DeepSeek V3.1 | 0.94 | 0.25 | 0.01 | 0.00 | 0.95 | 0.95 | 0.34 |
| Gemini 2.5 Flash | 0.58 | 0.11 | 0.00 | 0.00 | 0.99 | 0.96 | 0.33 |
| GPT-5 | 0.74 | 0.09 | 0.00 | 0.00 | 0.98 | 0.92 | 0.66 |
| Qwen3-235B | 0.94 | 0.62 | 0.05 | 0.01 | 0.99 | 0.98 | 0.28 |

Table 9: Exact-match accuracy on direct multi-digit arithmetic; columns indicate digits per operand. Each entry averages 100 IID problems per setting with uniformly sampled $d$-digit operands. Two-shot prompting, single-integer output only.

**Results.** Even the most recently released LLMs still struggle with *multi-digit multiplication*, while fail to achieve perfect accuracy on *addition*. For multiplication, accuracy drops sharply as operand length increases: models perform well on 3-digit multiplication ($\geq 0.58$), but fall below $0.10$ for 5-digit cases and nearly 0 at 6 digits. Qwen3-235B is the most robust, reaching $0.62$ on 4-digit and $0.05$ on 5-digit multiplication, yet still fails on 6 digits. For addition, all models achieve $\geq 0.95$ on 7–8 digit tasks, but accuracy declines on long-sequence addition (50 digits), ranging from $0.28$ (Qwen3-235B) to $0.82$ (Claude 3.7 Sonnet). In summary, LLMs excel at addition with short to medium operands but remain brittle for both long-sequence addition and especially large-digit multiplication.

## E.2  EMERGENCE OF FOURIER FEATURES DURING PRE-TRAINING

We follow Zhou et al. (2024) and conduct the same Fourier analysis on Pythia model. In Figure 9, we show how Pythia gradually learns the Fourier features during pre-training. With different model size, the model gradually learn the same frequency components.

We extend the work of Zhou et al. (2024) to other pre-trained LLMs and observe similar findings: pre-trained LLMs, regardless of the dataset used, tend to learn the same outlier frequency components.

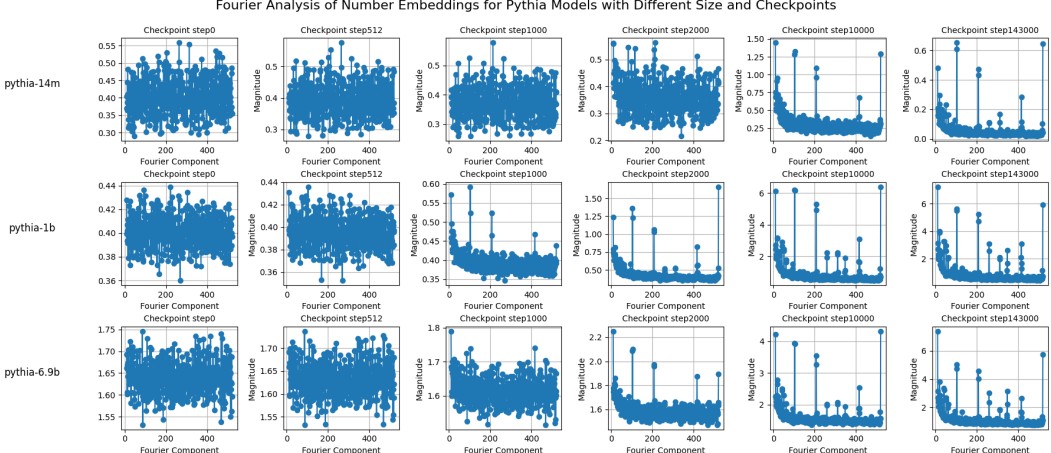

Figure 9: Fourier analysis of the Pythia model's number embeddings across pre-training checkpoints. The figure illustrates how the Fourier features are progressively learned during pre-training, showing the emergence of specific frequency components. Models of varying sizes exhibit a similar trend, gradually learning the same frequency components over time.

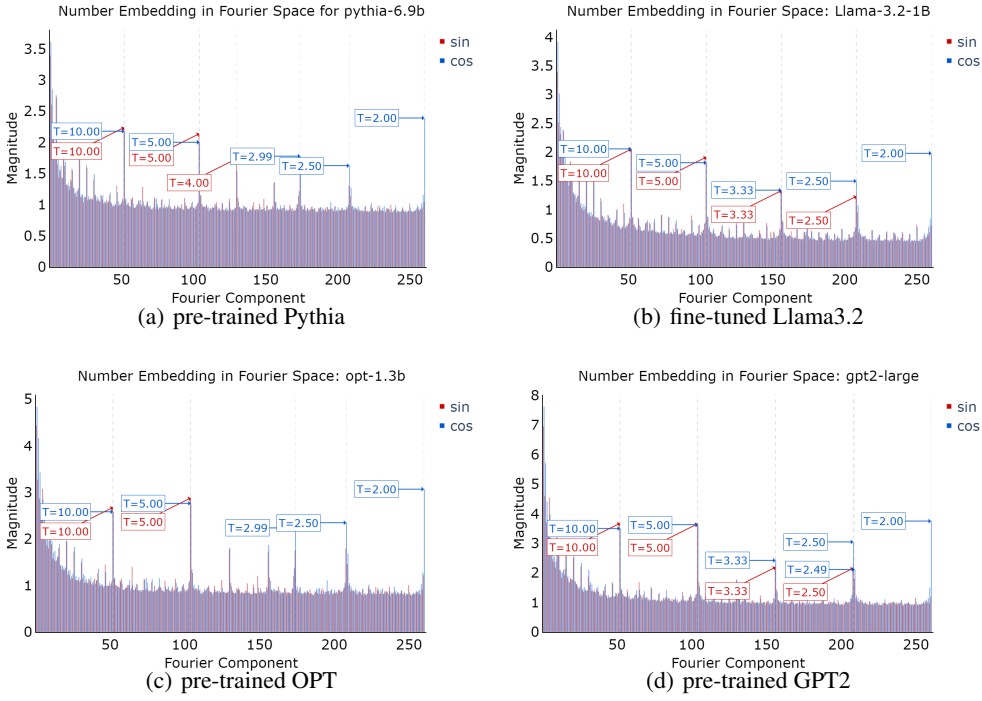

Figure 10: Number embedding in Fourier space for different pre-trained models.

# F   FONE FOR 60-DIGIT INTEGER ADDITION IN ONE FORWARD PASS

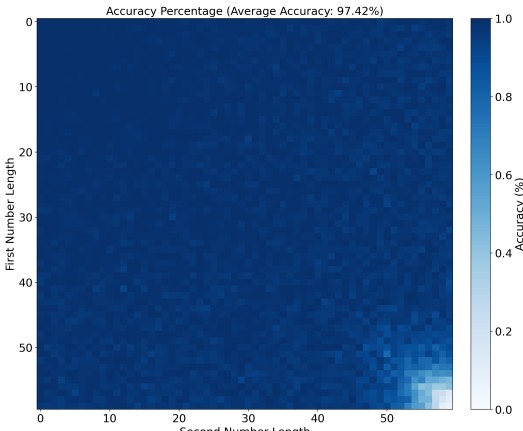

Figure 11: Accuracy of an 8-layer transformer on 60-digit addition tasks, illustrating the effectiveness of FoNE embeddings in handling long sequences. The model achieves an average accuracy of 97.42% across different operand lengths, showcasing its capability in numerical precision and sequence representation.

As discussed in Section 5, the maximum digit length that a `float64` data type can precisely represent is 15 digits. Consequently, even if we convert numbers to `float64` and then back to `float16` to match the model weight precisionm it still introduce numerical inaccuracies when the input $x$ exceeds 15 digits. To mitigate this issue, we process $x$ by dividing it into smaller chunks, allowing the FoNE to operate effectively without precision loss.

Specifically, $x$ is split into groups of five digits, and FoNE is applied independently to each chunk. Each digit within a chunk is encoded into two dimensions, resulting in an embedding of length 10 per chunk. These chunk embeddings are then concatenated to form the final representation of $x$. This method ensures that even for long inputs, the FoNE still preserve the numeracy of the numbers.

We adopt the data generation approach from McLeish et al. (2024a), which includes all combinations of operand lengths $(i, j)$ up to a maximum length $k$, generating 20 million stratified samples to ensure balanced representation across all length pairs. Training is conducted using a language model cramming approach (Geiping & Goldstein, 2023), constrained to 8 exaFLOP (equivalent to 24 hours of training on a single Nvidia RTX A6000 GPU). Using this strategy, as depicted in Figure 4(a), an 8-layer transformer trained on 60-digit addition achieves an average accuracy of 97.42% across various operand lengths in just one forward pass. This result underscores the effectiveness of the FoNE in processing long numbers with high precision and computational efficiency in just one forward pass.

# G   COMBINING FONE WITH ABACUS

We train decoder-only causal language models to solve arithmetic problems, following the setup described in McLeish et al. (2024a). Inputs are formatted in a least-significant-digit-first order (e.g., $98282 + 3859172 = 2787472$), without padding between digits or operands. The training dataset includes all combinations of operand lengths $(i, j)$ up to a maximum length $k$, with 20 million stratified samples ensuring balanced representation across all length pairs.

For input representation, we combine Fourier Number Embeddings (FoNE) with the Abacus method McLeish et al. (2024a). That each digit is embedded with FoNE. Training is conducted using a language model cramming approach (Geiping & Goldstein, 2023), constrained to 8 exaFLOP (equivalent to 24 hours of training on a single Nvidia RTX A6000 GPU).

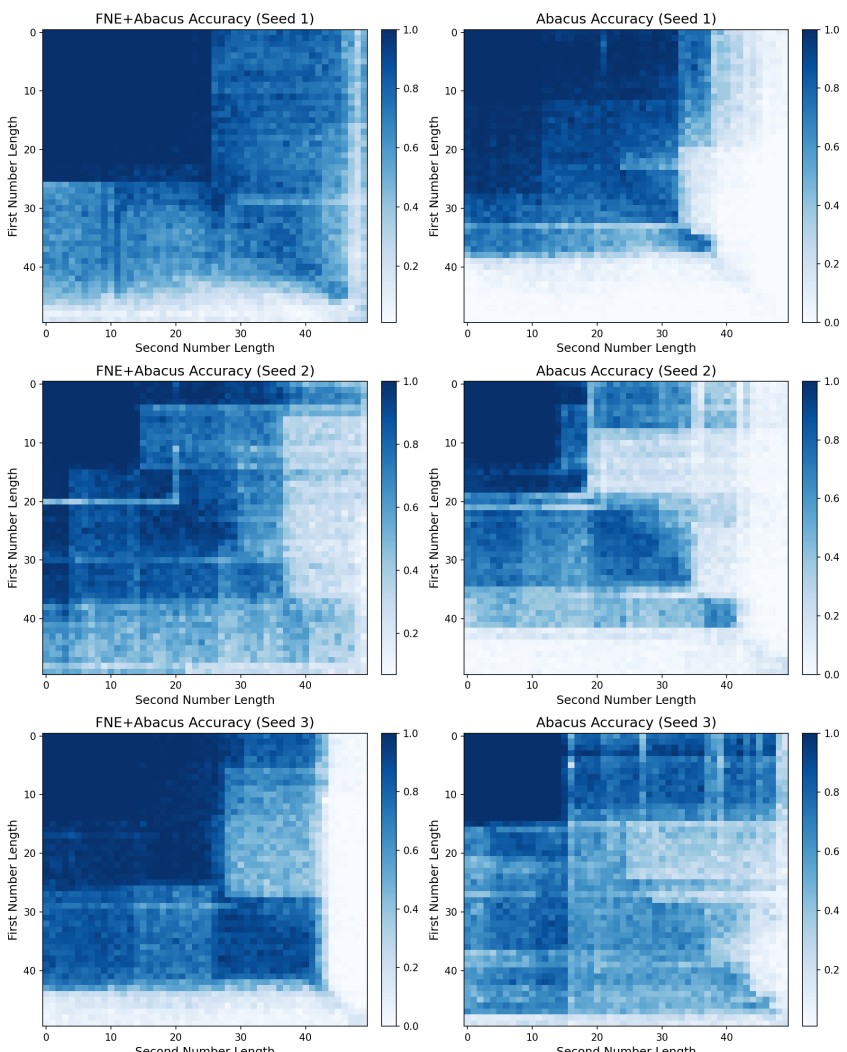

Figure 12: Heatmaps of accuracy percentages for "FoNE+Abacus" (left column) and "Abacus" (right column) across three different random seeds. Each heatmap represents accuracy as a function of the first and second number lengths, with lighter blue shades indicating higher accuracy. The color scale ranges from white (low accuracy) to blue (high accuracy). These visualizations highlight FoNE can combine with Abacus to improve performance.

We train and evaluate the models across three runs, each with a different random seed, as shown in Figure 12. Results indicate that incorporating FoNE enables the Abacus method to achieve better generalization and higher accuracy.

## H   MORE DETAILS ON EXPERIMENTAL SETUP

In this section, we provide the experiments settings that we used in the Section 4.1.

Learning rates were determined through an extensive search, with the best rates selected separately for each method based on validation performance. Final training hyperparameters include a learning rate of $0.005$ for regular and FoNE methods, and $0.0001$ for the xVal method, a batch size of $512$, and $100$ epochs. The fine-tuning process required less than 10 hours, while training from scratch took less than 3 days.

| Dataset | Train Size | Validation Size | Test Size |
|---|---|---|---|
| 6-digit decimal addition | 720,000 | 80,000 | 200,000 |
| 6-digit integer addition | 720,000 | 80,000 | 200,000 |
| 5-digit integer subtract | 720,000 | 80,000 | 200,000 |
| 3-digit integer multiplication | 360,000 | 40,000 | 100,000 |
| 4-digit integer multiplication | 720,000 | 80,000 | 200,000 |
| classification | 720,00 | 80,00 | 200,00 |

Table 10: Dataset Sizes for Training, Testing, and Validation

| Dataset | Model Size for Varying Data Size | Data Size for Varying Model Size |
|---|---|---|
| 6-digit decimal addition | 37.55M | 200,000 |
| 6-digit integer addition | 37.55M | 200,000 |
| 5-digit integer subtract | 37.55M | 200,000 |
| 3-digit integer multiplication | 37.55M | 360,000 |
| 4-digit integer multiplication | 37.55M | 360,000 |
| 4-digit integer multiplication | 37.55M | 360,000 |
| classification | 37.55M | 50,000 |

Table 11: Dataset and Configuration Sizes for Model and Data Variation Experiments

### H.1 ABLATION STUDY

In this section, we present the mispredictions of the model trained with an FoNE, where the periods are multiples of $5$ instead of $10$. Table 13 demonstrates that, for each digit, the mispredictions consistently deviate from the true labels by $5$.

We also present the model's mispredictions in Table 14, where each digit is encoded into a separate dimension of the embedding. For example, the number $567$ is represented as $[5, 6, 7]$. During training, we compute the RMSE loss between the last hidden states and the labels. During prediction, we interpret each entry in the last hidden state as a single digit.

| Model | Hidden Size | Intermediate Size | # Hidden Layers | # Attention Heads | # Key-Value Heads |
|---|---|---|---|---|---|
| 1 | 64 | 256 | 1 | 4 | 2 |
| 2 | 128 | 512 | 2 | 4 | 2 |
| 3 | 192 | 768 | 3 | 6 | 3 |
| 4 | 256 | 1024 | 4 | 8 | 4 |
| 5 | 320 | 1280 | 5 | 8 | 4 |
| 6 | 384 | 1536 | 6 | 8 | 4 |

Table 12: Model Configuration Table

Table 13: Mispredictions in the Final Evaluation with when we embed each digit with only $\mathrm{mod}5$.

| Index | Predicted Value | Actual Value |
|---|---|---|
| 1 | 934.03 | 934.585 |
| 2 | 3.009 | 558.509 |
| 3 | 912.311 | 917.366 |
| 4 | 6201.003 | 1756.008 |
| 5 | 1240.34 | 1290.84 |

# I    REPLICATING RESULTS ON GPT2-LARGE BASED MODEL

We conduct the same experiments on decimal addition using a GPT-2 Large-based model. The results indicate that changing the model architecture does not affect the outcomes. For instance, GPT-2 Large employs LayerNorm, while Llama 3.2 uses RMSNorm.

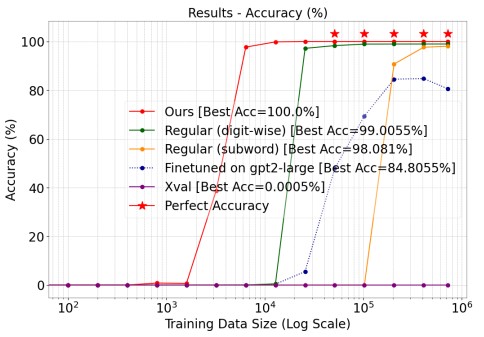
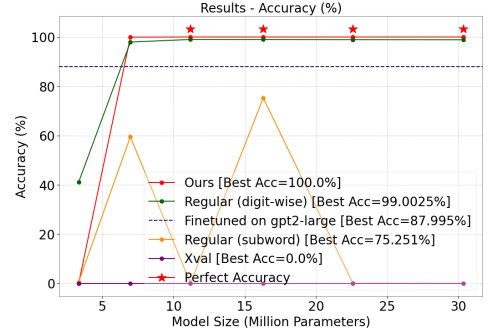

(a) 6-digit decimal addition: Accuracy vs. Training Data Size

(b) 6-digit decimal addition: Accuracy vs. Model Size

Figure 13: We train GPT2-Large from scratch with random initialization using different number embedding methods on 6-digit decimal addition. The test accuracy is compared across varying data sizes and model sizes.

Table 14: Mispredictions in the Final Evaluation when directly encoding numbers into their embeddings.

| Index | Predicted Value | Actual Value |
|-------|-----------------|--------------|
| 1 | 883.888 | 993.999 |
| 2 | 787.878 | 898.989 |
| 3 | 888.758 | 989.759 |
| 4 | 748.785 | 849.895 |
| 5 | 677.677 | 688.788 |
| 10 | 1179.488 | 1189.499 |

## J $R^2$ COMPARISON FOR DIFFERENT ARITHMETIC TASKS

xVal Golkar et al. (2023) performs well on the $R^2$ metric

$$R^2 = 1 - \frac{\sum_{i=1}^{n}(y_i - \widehat{y}_i)^2}{\sum_{i=1}^{n}(y_i - \bar{y})^2},$$

because it uses RMSE as its loss function. However, we demonstrate that FoNE outperforms xVal on $R^2$ in most tasks. We show the final $R^2$ on test dataset in our experiments(Section 4.2).

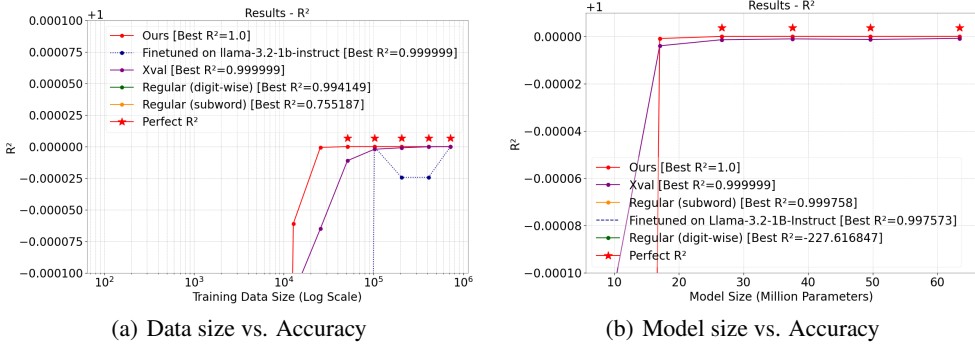

(a) Data size vs. Accuracy  (b) Model size vs. Accuracy

Figure 14: Comparison of $R^2$ trends for 6-digit decimal addition with respect to model size and data size.

## K COMPARISON WITH NUMBER TOKEN LOSS (NTL)

To validate the effectiveness of FoNE, we conduct comprehensive experiments comparing against Number Token Loss (NTL) (Zausinger et al., 2024), a method designed to improve numerical reasoning by incorporating new losses between number tokens. We evaluate two NTL variants: NTL with digit-wise tokenization (denoted NTL-D), where numbers are decomposed into individual digit tokens, and NTL with subword tokenization (NTL-S), using the standard tokenizer with an extended number vocabulary covering tokens 0-999. For each variant, we test three lambda values ($\lambda \in \{0.15, 0.3, 0.8\}$) that control the relative weighting of the NTL loss component.

### K.1 EXPERIMENTAL SETUP

We conduct experiments on two arithmetic reasoning benchmarks: 4-digit integer addition and 3-digit integer multiplication. To assess data efficiency, we train models with varying amounts of data: 5K, 25K, and 50K samples for addition; 10K, 50K, and 75K samples for multiplication. All experiments use Llama-3.2-1B-Instruct's configuration, training from scratch with consistent architectural parameters (512 hidden dimensions, 8 layers, 8 attention heads) to ensure fair comparison. We train all models for 50 epochs with batch size 512, using learning rates of $5 \times 10^{-4}$ for FoNE and $5 \times 10^{-3}$ for NTL methods based on preliminary tuning. Models are evaluated on 1000 held-out test samples using whole number accuracy, $R^2$, and RMSE.

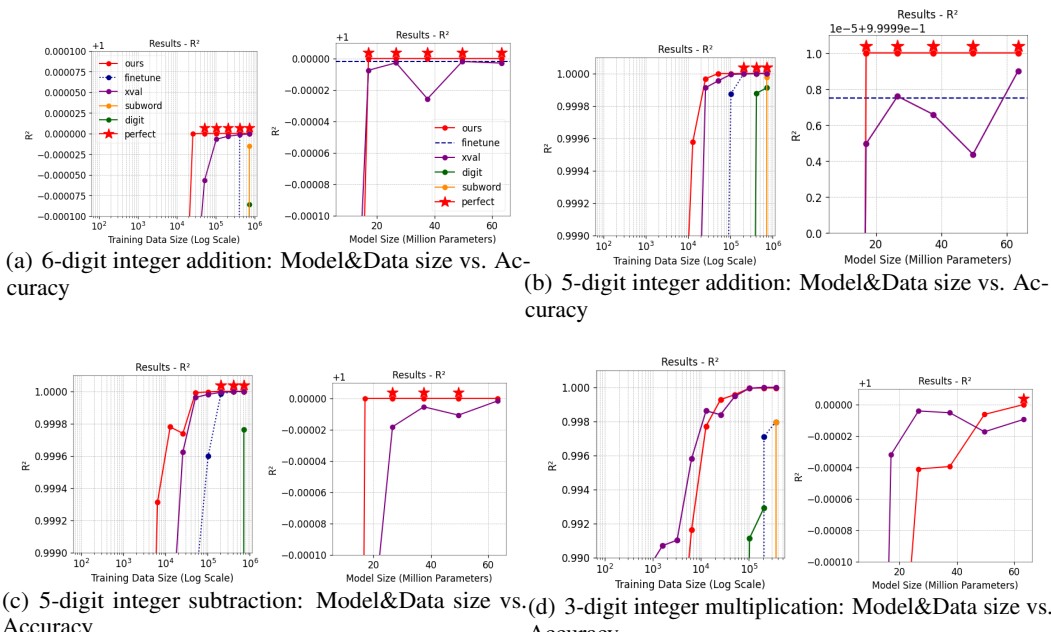

Figure 15: Comparison of $R^2$ trends for various arithmetic tasks with respect to model size and data size.

## K.2 RESULTS

Figure 16 presents comprehensive performance comparisons across both tasks and all metrics. The results demonstrate FoNE's substantial and consistent superiority over all NTL variants.

**Addition Task** FoNE reach 98.6% accuracy with only 5,000 training samples and perfect 100% accuracy with 25,000 or more samples (Table 15). The $R^2$ scores consistently exceed 0.999, and RMSE drops to effectively zero at larger training scales. NTL methods exhibit significantly degraded performance. The best-performing variant, NTL(D,0.15), achieves 97.4% accuracy with 50K samples—still below FoNE's performance at 5K samples. Other NTL configurations struggle considerably: NTL(D,0.3) peaks at only 51.4% accuracy, while higher lambda values ($\lambda = 0.8$) yield near-zero performance (3.3-6.1%). Most critically, NTL with subword tokenization fails catastrophically across all configurations, achieving near-zero accuracy with negative $R^2$ values as low as -13,170, indicating performance substantially worse than a naive baseline. RMSE values for NTL(S) exceed $4.7 \times 10^5$, compared to 0.0 for FoNE.

**Multiplication Task** The multiplication task proves more challenging, yet FoNE maintains strong performance. With 75,000 training samples, FoNE achieves 89.3% accuracy with $R^2$=0.9996 and RMSE=4,286, demonstrating consistent and reliable predictions (Table 16). Performance scales smoothly with data: from 35.1% at 10K samples to 84.5% at 50K and 89.3% at 75K. NTL methods again significantly underperform. The best NTL configuration, NTL(D,0.15) with 75K samples, achieves only 16.3% accuracy—73 percentage points below FoNE. Most other NTL variants achieve less than 5% accuracy even at the largest training scale. Subword tokenization continues to fail, with negative $R^2$ scores (e.g., -0.534 for NTL(S,0.8)) and RMSE values exceeding $2 \times 10^5$, three orders of magnitude worse than FoNE.

**Analysis** Several key observations emerge from our experiments. First, FoNE demonstrates superior data efficiency, achieving strong performance with minimal training data while NTL requires substantially more samples to reach even moderate accuracy levels. Second, NTL's performance is highly sensitive to the lambda hyperparameter: increasing $\lambda$ from 0.15 to 0.8 consistently degrades accuracy across both tasks, suggesting that heavily weighting the distance-based loss interferes with

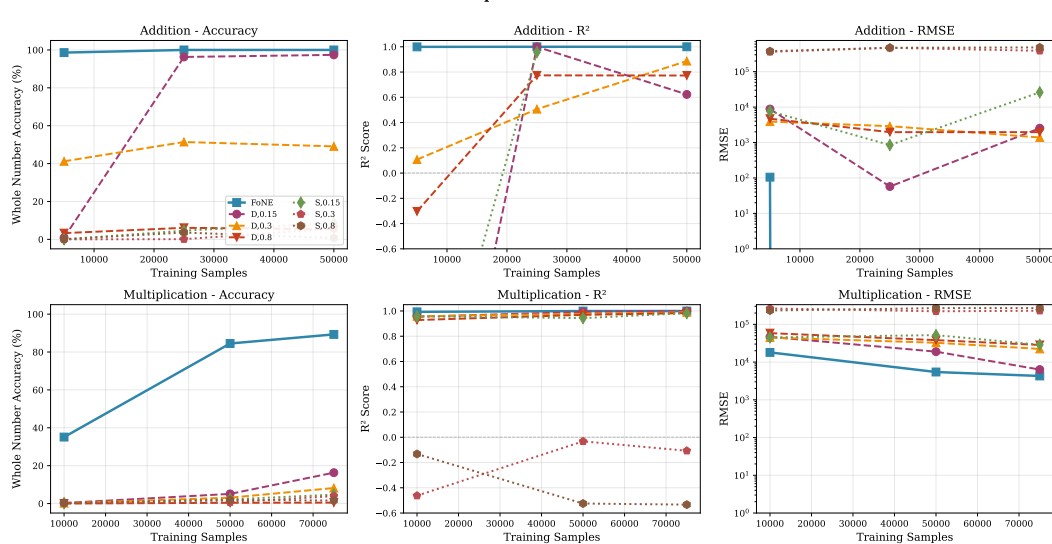

Figure 16: Performance comparison of FoNE vs. NTL methods on addition (top) and multiplication (bottom) tasks across three metrics: accuracy, $R^2$, and RMSE. FoNE (solid blue squares) consistently outperforms all NTL variants, achieving 100% accuracy on addition and 89.3% on multiplication vs. NTL's best of 16.3%.

Table 15: Addition task results comparing FoNE with NTL methods across different training data sizes. FoNE achieves perfect accuracy with $\geq$25K samples while all NTL variants struggle.

| Method | 5K Samples | | 25K Samples | | 50K Samples | |
|---|---|---|---|---|---|---|
| | Acc. (%) | $R^2$ | Acc. (%) | $R^2$ | Acc. (%) | $R^2$ |
| **FoNE** | **98.6** | **0.9993** | **100.0** | **1.0000** | **100.0** | **1.0000** |
| NTL(D,0.15) | 0.9 | -3.57 | 96.3 | 0.9998 | 97.4 | 0.623 |
| NTL(D,0.3) | 41.2 | 0.107 | 51.4 | 0.506 | 49.1 | 0.887 |
| NTL(D,0.8) | 3.3 | -0.304 | 6.1 | 0.774 | 5.3 | 0.773 |
| NTL(S,0.15) | 0.0 | -2.41 | 4.6 | 0.957 | 8.6 | -39.3 |
| NTL(S,0.3) | 0.0 | -7763 | 0.1 | -13030 | 5.7 | -8884 |
| NTL(S,0.8) | 0.1 | -8418 | 3.6 | -13170 | 0.7 | -13730 |

learning. Finally, FoNE also exhibits better training efficiency, requiring $2.5 - 11.7\times$ less training time than NTL methods while achieving far superior accuracy.

Table 16: Multiplication task results. FoNE maintains strong performance (89.3% at 75K samples) while the best NTL variant achieves only 16.3%.

| Method | 10K Samples | | 50K Samples | | 75K Samples | |
|---|---|---|---|---|---|---|
| | Acc. (%) | $R^2$ | Acc. (%) | $R^2$ | Acc. (%) | $R^2$ |
| **FoNE** | **35.1** | **0.9932** | **84.5** | **0.9994** | **89.3** | **0.9996** |
| NTL(D,0.15) | 0.3 | 0.955 | 5.1 | 0.993 | 16.3 | 0.9991 |
| NTL(D,0.3) | 0.2 | 0.958 | 3.1 | 0.978 | 8.2 | 0.990 |
| NTL(D,0.8) | 0.0 | 0.928 | 0.4 | 0.969 | 0.5 | 0.983 |
| NTL(S,0.15) | 0.2 | 0.957 | 2.2 | 0.944 | 4.7 | 0.983 |
| NTL(S,0.3) | 0.2 | -0.462 | 0.7 | -0.032 | 3.9 | -0.107 |
| NTL(S,0.8) | 0.3 | -0.132 | 1.9 | -0.525 | 1.6 | -0.534 |

THE USE OF LLMS

LLMs were used only to polish language, such as grammar and wording. These models did not contribute to idea creation or writing, and the authors take full responsibility for this paper's content.

