# OpenReview forum: "FoNE: Precise Single-Token Number Embeddings via Fourier Features"
_ICLR.cc/2026/Conference — ICLR 2026 Poster_

### Official Review · Reviewer_pYSK · 2025-10-27

**Soundness:** 2
**Presentation:** 3
**Contribution:** 3
**Rating:** 2
**Confidence:** 4

**Summary:**

The paper presents a new number encoding for LLMs based on Fourier features. The embedding is the sum of a number token and the separately constructed Fourier number embedding (FoNE). Experiments on synthetic datasets with addition and multiplication tasks show the superiority of the method over other number representations such as a simple single-digit-encoding or the xval approach.

**Strengths:**

1. The approach is novel and makes sense. Using Fourier features is intuitively more suitable for numerical reasoning than standard number tokenization, and the encoding-decoding approach is elegant.
2. The paper is well written and the methodology is easy to follow.

**Weaknesses:**

1. The experiment is not sufficient to show the superiority of the approach. Why is the method not tested on standard benchmarks for mathematical reasoning, such as GSM8K or MATH? The synthetic datasets is very artificial since it only contains equations of fixed-digit numbers, no other text. However, number representations in LLMs are only relevant for data that consists of both text and numerical tasks.
2. The method seems incompatible with existing models since it requires specific modifications to the tokenizer and an edit after the embedding, so it is not applicable for finetuning, only for training models from scratch. This makes it difficult to evaluate the effectiveness, and limits the applicability of the method.
3. The method is only compared to xval as a baseline approach, although there have been other papers on this topic, such as the Number Token Loss (NTL) [1]. In the discussion section, it is argued that NTL is not relevant since "regression produces continuous value", but NTL is only a loss function that does not impose any constraints on what the number looks like. It should be comparable. In general, the paper only reports accuracy as a metric, although the numeric value clearly matters and regression metrics such as RMSE would be suitable.

[1] Zausinger, Jonas, et al. "Regress, Don't Guess--A Regression-like Loss on Number Tokens for Language Models." arXiv preprint arXiv:2411.02083 (2024).

**Questions:**

1. The main question is how the method compares to a simple transformer model on a standard numerical reasoning dataset.
2. m and n are the number of integer and decimal digits, but m and n seem to be set for each number, and not by the maximum m and n of the dataset. Doesn’t this lead to issues because the resulting FoNE vectors are not aligned, I.e. the third cell for 1000 would represent the tens digit but the third cell for 0.12 would be one  hundredths?
3. In definition 3.6, why is it \phi(0, 10) to \phi(9,10)? I expected it to be dependent on i and not fixed to 10.
4. How efficient is the end to end training? The manuscript claims that this method is even more efficient than the baselines since fewer tokens are used, but I imagine that computing the FoNE representation and adding it to the embedding vector creates an overhead. How long does training one epoch take in comparison to the baselines?

---

> ### Author Response · Authors · 2025-11-22
> **Rebuttal by Authors**
>
> We are glad the reviewer finds our method novel and elegant. We have carefully addressed your concerns and hope you will consider raising your score in light of our response.
>
>
> ---
>
>
> **Q1: The experiment is not sufficient to show the superiority of the approach. Why is the method not tested on standard benchmarks for mathematical reasoning. The synthetic dataset is very artificial since it only contains equations of fixed-digit numbers, no other text. However, number representations in LLMs are only relevant for data that consists of both text and numerical tasks.**
>
>
> A1. Following previous work [1,2,3,4], we use synthetic benchmarks intentionally to isolate the fundamental properties of numeracy-preserving embeddings: (1) strong generalization to unseen numbers (Figure 1: 99% accuracy on 6-digit addition with only 6K examples—100× more data-efficient than baselines), and (2) stable representation of very large numbers (Figure 4: 60-digit addition). These properties cannot be reliably measured in noisy real-world benchmarks where multiple confounding factors exist.
>
>
> **Real-world validation:** Regarding GSM8K/MATH, these benchmarks require: (1) large-scale pretraining on diverse corpora, (2) instruction fine-tuning, and (3) chain-of-thought training—requiring computational resources beyond our scope.  Hence, we train GPT-2-117M from scratch on 10B tokens from FineWeb (data that consists of both text and numerical tasks) with different tokenization schemes:BPE, Single-Digit, xVal, and FoNE.
>
> **Experiment Results:**
>
> Perplexity on FineWeb validation set:
>
> | Method | Validation Perplexity |
> |--------|----------------------|
> | FoNE | **46.8** |
> | xVal | 48.8 |
> | Single-Digit | 48.7 |
> | BPE | 52.6 |
>
>
>
> Perplexity of text tokens only on sequences that contain both text and numbers. ($4,984,914$ tokens in total).:
>
> | Method | Validation Perplexity |
> |--------|----------------------|
> | FoNE | **46.856** |
> | xVal | 49.160 |
> | Single-Digit | 55.922 |
> | BPE | 67.390 |
>
>
> FoNE model achieves the lowest validation perplexity on the FineWeb validation set. Furthermore, we evaluate the perplexity of text tokens only on sequences that contain both text and numbers. FoNE remains the lowest perplexity while other methods show even worse perplexity compared to their performance on all sequences.
>
> Since numbers in FineWeb appear naturally interleaved with text, this experiment shows that using FoNE throughout pretraining not only preserves the model's semantic abilities but can in fact **improve** its language modeling quality by giving better number representation.
>
>
> **Overall, our synthetic experiments establish the foundation and our real-world pretraining confirms practical applicability.**
>
>
> [1] McLeish, Sean, et al. "Transformers can do arithmetic with the right embeddings." Advances in Neural Information Processing Systems 37 (2024): 108012-108041.
>
> [2] Golkar, Siavash, et al. "xval: A continuous number encoding for large language models." arXiv preprint arXiv:2310.02989 (2023).
>
> [3] Thawani, Avijit, et al. "Representing numbers in NLP: a survey and a vision." Proceedings of the 2021 conference of the North American chapter of the association for computational linguistics: human language technologies. 2021.
>
> [4] Lee, Nayoung, et al. "Teaching arithmetic to small transformers." arXiv preprint arXiv:2307.03381 (2023).

---

> > ### Author Response · Authors · 2025-11-22
> > **Rebuttal by Authors - NTL Experiments**
> >
> > ## Additional Experiments: Comparison with Number Token Loss (NTL)
> >
> > Following the reviewer's suggestion, we have conducted comprehensive experiments comparing FoNE with NTL using their exact experimental setup. We train a 38M Transformer model on multiplication and addition tasks with varying training data sizes and report all metrics requested by the reviewer. (See full experiments in Section K)
> >
> > ### Results Summary
> >
> > **Table: Multiplication Performance Comparison**
> >
> > #### Whole Number Accuracy (%)
> >
> > | Training Samples | FoNE | NTL(D,0.15) | NTL(D,0.3) | NTL(D,0.8) | NTL(S,0.15) | NTL(S,0.3) | NTL(S,0.8) |
> > |------------------|------|-------------|------------|------------|-------------|------------|------------|
> > | 10,000 | **35.1** | 0.3 | 0.2 | 0.0 | 0.2 | 0.2 | 0.3 |
> > | 50,000 | **84.5** | 5.1 | 3.1 | 0.4 | 2.2 | 0.7 | 1.9 |
> > | 75,000 | **89.3** | 16.3 | 8.2 | 0.5 | 4.7 | 3.9 | 1.6 |
> >
> > #### $R^2$ Score
> >
> > | Training Samples | FoNE | NTL(D,0.15) | NTL(D,0.3) | NTL(D,0.8) | NTL(S,0.15) | NTL(S,0.3) | NTL(S,0.8) |
> > |------------------|------|-------------|------------|------------|-------------|------------|------------|
> > | 10,000 | **0.9931** | 0.9546 | 0.9582 | 0.9282 | 0.9575 | -0.4624 | -0.1324 |
> > | 50,000 | **0.9994** | 0.9925 | 0.9777 | 0.9694 | 0.9436 | -0.0324 | -0.5248 |
> > | 75,000 | **0.9996** | 0.9992 | 0.9895 | 0.9830 | 0.9826 | -0.1069 | -0.5336 |
> >
> > #### RMSE (Root Mean Squared Error, lower is better)
> >
> > | Training Samples | FoNE | NTL(D,0.15) | NTL(D,0.3) | NTL(D,0.8) | NTL(S,0.15) | NTL(S,0.3) | NTL(S,0.8) |
> > |------------------|------|-------------|------------|------------|-------------|------------|------------|
> > | 10,000 | **18,066** | 46,532 | 44,649 | 58,489 | 45,024 | 2.6e+05 | 2.3e+05 |
> > | 50,000 | **5,458** | 18,901 | 32,567 | 38,185 | 51,832 | 2.2e+05 | 2.7e+05 |
> > | 75,000 | **4,287** | 6,333 | 22,323 | 28,467 | 28,802 | 2.3e+05 | 2.7e+05 |
> >
> >
> > **Legend:**
> > - **NTL(D,λ)**: Number Token Loss with digit-wise tokenization, $\lambda$ = weighting hyperparameter
> > - **NTL(S,λ)**: Number Token Loss with subword tokenization, $\lambda$ = weighting hyperparameter
> > - Bold values indicate best performance in each row

---

> > > ### Comment · Reviewer_pYSK · 2025-11-27
> > >
> > > Thank you for the thorough response. Many of my questions were answered and resolved, including Q3 (comparison to number token loss) which was now added, Q4 (RMSE), Q5 (setting m and n), and Q6 (definition 3.6). The new experiments on FineWeb and Number token loss significantly improve the paper.
> > >
> > > However, my main concern remains and is shared by the other reviewers: The method is not tested on standard benchmarks for arithmetics and numerical reasoning. I believe the reason is basically due to my Q2; namely that the method is not a plug-in method, and requires to retrain models from scratch (which requires more computational resources). This limitation is not an issue in general, but becomes an issue if it means that the necessary experiments can not be conducted.
> > >
> > > It also impacts the comparison to baselines, because (1) NTL was trained from scratch as well, although it is a method that is compatible with pretrained model, so the argument that FoNE has a better sample efficiency than NTL does not really count, (2) the runtime comparison (my Q7) was conducted on the toy dataset, which consists only of numbers, where FoNE has a huge advantage due to the smaller number of tokens required. I'm still not convinced that the training is more efficient when training on a large corpus of mixed text and numbers.
> > >
> > > I raise my score to 4 since the additional experiments show benefits of the method, but I still feel that important parts are missing.

---

> > > > ### Author Response · Authors · 2025-12-03
> > > > **Thanks for raising your score**
> > > >
> > > > We are glad that our new experiments on FineWeb and NTL **significantly improve the paper**.
> > > >
> > > > ---
> > > >
> > > > **Q: The method is not a plug-in method and not tested on standard benchmarks for arithmetic and numerical reasoning**
> > > >
> > > > A:  Regarding your remaining concern, in this paper, our primary goal is to highlight the importance of number embeddings and representations, following prior work [1,2,3,4]. Although we do not aim to develop a plug-in method for pretrained models, simply aligning a pretrained model’s number embeddings to FoNE already yields improvements on standard benchmarks.
> > > >
> > > > A simple alternative approach is as follows:
> > > > take the pretrained model’s number embeddings, amplify the Fourier components used in FoNE, and then finetune the model on downstream tasks.
> > > >
> > > > To demonstrate the effectiveness of our method, we finetune and evaluate Llama-3.2-3B with LoRA on the GSM8K-Range [5] benchmark, which contains three difficulty levels of GSM8K questions, with three random seeds for each level.
> > > >
> > > > Table 1: Limited Data (11,700 training samples)
> > > >
> > > > | Level | Baseline (%)    | Amplify 1.2x (%) | Improvement (%) | Rel Gain (%) |
> > > > |-------|-----------------|------------------|-----------------|--------------|
> > > > | 4     | 26.69 ± 0.60    | 29.13 ± 0.69     | +2.44           | **+9.2**        |
> > > > | 5     | 22.66 ± 0.83    | 22.87 ± 0.31     | +0.20           | **+0.9**         |
> > > > | 6     | 17.62 ± 0.41    | 19.04 ± 0.42     | +1.42           | **+8.1**         |
> > > >
> > > > Table 2: Full Data (26,400 training samples)
> > > >
> > > > | Level | Baseline (%)    | Amplify 1.2x (%) | Improvement (%) | Rel Gain (%) |
> > > > |-------|-----------------|------------------|-----------------|--------------|
> > > > | 4     | 42.69 ± 0.59    | 43.47 ± 0.53     | +0.78           | **+1.8**         |
> > > > | 5     | 36.90 ± 1.21    | 37.63 ± 0.46     | +0.73           | **+2.0**         |
> > > > | 6     | 33.70 ± 1.03    | 34.36 ± 0.56     | +0.66           | **+1.9**         |
> > > >
> > > >
> > > >
> > > >
> > > > Even with this very simple modification—aligning the pretrained number embeddings to FoNE—**the model improves. Notably, the gains are larger when training data is limited, consistent with our findings on synthetic data**.
> > > >
> > > >
> > > >
> > > > [1] McLeish, Sean, et al. "Transformers can do arithmetic with the right embeddings." Advances in Neural Information Processing Systems 37 (2024): 108012-108041.
> > > >
> > > >
> > > > [2] Golkar, Siavash, et al. "xval: A continuous number encoding for large language models." arXiv preprint arXiv:2310.02989 (2023).
> > > >
> > > >
> > > > [3] Thawani, Avijit, et al. "Representing numbers in NLP: a survey and a vision." Proceedings of the 2021 conference of the North American chapter of the association for computational linguistics: human language technologies. 2021.
> > > >
> > > >
> > > > [4] Lee, Nayoung, et al. "Teaching arithmetic to small transformers." arXiv preprint arXiv:2307.03381 (2023).
> > > >
> > > >
> > > >
> > > > [5] Shrestha, Safal, Minwu Kim, and Keith Ross. "Mathematical reasoning in large language models: Assessing logical and arithmetic errors across wide numerical ranges." arXiv preprint arXiv:2502.08680 (2025).

---

> ### Author Response · Authors · 2025-11-22
> **Rebuttal by Authors - Part 2**
>
> **Q2: The method seems incompatible with existing models since it requires specific modifications to the tokenizer and an edit after the embedding, so it is not applicable for finetuning, only for training models from scratch. This makes it difficult to evaluate the effectiveness, and limits the applicability of the method.**
>
> A2. Similar to other research about tokenization [1,2], FoNE does require a small tokenizer update and an additive embedding, and thus cannot be applied directly to already-pretrained models. However, this modification is necessary. Recent works [3,4,5] show that LLMs exhibit strong frequency bias on digits and numbers—models tend to output values that appear frequently in pretraining rather than numerically correct ones—which comes from pretraining. This bias is appropriate for text but actively harmful for numeracy, indicating that numbers must be treated as a separate modality rather than ordinary tokens.
>
> FoNE is designed specifically for pretraining, where numerical and textual signals can be learned together. Our small-scale pretraining experiment (A1) shows that FoNE improves language modeling perplexity, demonstrating that numeracy-preserving embeddings integrate cleanly with standard language learning.
>
> While FoNE is not a drop-in finetuning method, it is lightweight, compatible with standard pretraining pipelines, and intended to improve future model foundations.
>
> [1] Thawani, Avijit, et al. "Learn your tokens: Word-pooled tokenization for language modeling." arXiv preprint arXiv:2310.11628 (2023).
>
> [2] Tay, Yi, et al. "Charformer: Fast character transformers via gradient-based subword tokenization." arXiv preprint arXiv:2106.12672 (2021).
>
>
>
> [3] Shrestha, Safal, Minwu Kim, and Keith Ross. "Mathematical reasoning in large language models: Assessing logical and arithmetic errors across wide numerical ranges." arXiv preprint arXiv:2502.08680 (2025).
>
> [4] Razeghi, Yasaman, et al. "Impact of pretraining term frequencies on few-shot reasoning." arXiv preprint arXiv:2202.07206 (2022).
>
> [5] Mirzadeh, Iman, et al. "Gsm-symbolic: Understanding the limitations of mathematical reasoning in large language models." arXiv preprint arXiv:2410.05229 (2024).
>
> ---
>
> **Q3: The method is only compared to xval as a baseline approach, although there have been other papers on this topic, such as the Number Token Loss (NTL).**
>
>
>
> A3. Thank you for this valuable suggestion. We have now conducted comprehensive experiments comparing FoNE with NTL across multiple configurations, reporting accuracy, $R^2$ and RMSE. The new results are presented in Section K in the paper.
>
> NTL and FoNE address complementary aspects—NTL modifies the loss function while FoNE changes the input representation. Nevertheless, the comparison reveals important insights:
>
> (1) Low exact match accuracy for NTL: While NTL achieves comparable $R^2$ scores (0.9991 vs. 0.9996, slightly worse than FoNE), its exact match accuracy remains substantially lower (16.3% vs. 89.3% on multiplication with 75K samples). However, a single digit error invalidates the entire results. Applications like phone numbers, verification codes, or multi-step arithmetic require exact values, not approximate ones.
>
> (2) More data required for NTL: FoNE achieves 100% accuracy on addition with 25K samples, while NTL requires 50K+ samples to reach 97.4%. This more than 2× data efficiency advantage becomes critical when scaling to larger models or more complex tasks.
>
> (3) $R^2$ can be misleading for discrete tasks: NTL achieves $R^2$=0.9991 on multiplication, but this high regression score masks low exact accuracy (16.3%). For example, predicting 668 instead of 669 yields $R^2 \approx 0.999$ but 0% exact match. In LLM applications requiring exact values, FoNE's 89.3% exact accuracy is far more useful than NTL's approximate predictions, even when both show high $R^2$.
>
> ---
>
>
> **Q4. In general, the paper only reports accuracy as a metric, although the numeric value clearly matters and regression metrics such as RMSE would be suitable.**
>
> A4. In our paper, we did report regression evaluation metric $R^2$ scores for all tasks in Appendix J. We chose not to use RMSE as the primary metric because it produces very large values for most methods except FoNE, making cross-method comparisons difficult to interpret.

---

> > ### Author Response · Authors · 2025-11-22
> > **Rebuttal by Authors - Part 3**
> >
> > **Q5. m and n are the number of integer and decimal digits, but m and n seem to be set for each number, and not by the maximum m and n of the dataset.**
> >
> > A5. Thank you for raising this important clarification. To be clear: **m and n are fixed globally for the entire dataset, not set per number**. This ensures perfect alignment across all numbers.
> >
> > Specifically, FoNE embeds every number using the same fixed sequence of Fourier periods $\{10^{-n+1}, \ldots, 10^{m}\}$. Each dimension pair always corresponds to the same modulus $x \bmod 10^k$ for all numbers, regardless of their magnitude. For example, if $m=3$ and $n=2$, the periods are $\{10^{-1}, 10^{0}, 10^{1}, 10^{2}, 10^{3}\}$:
> > - The 2nd dimension pair always represents $x \bmod 10^{0} = x \bmod 1$ (the fractional part)
> > - For 1234: this captures $1234 \bmod 1 = 0$ (no fractional component)
> > - For 56.78: this captures $56.78 \bmod 1 = 0.78$ (the fractional part)
> > - Both use the same embedding dimension for the same scale
> > No misalignment occurs—the embeddings are structurally identical across all numbers, with each position representing a consistent scale. We have clarified this in Section 3.1 following Definition 3.1.
> >
> > ---
> >
> > **Q6. In definition 3.6, why is it $\phi(0, 10)$ to $\phi(9,10)$? I expected it to be dependent on i and not fixed to 10.**
> >
> > A6. This is because each digit position is decoded independently by comparing against all possible digit values (0-9), which always uses period $T=10$ regardless of position $i$.
> >
> > Example: For the number 42 with representation $[\cdots,\phi(42, 100), \phi(42, 10),\cdots]$:
> >
> > To decode the tens digit from $\phi(42, 100)$:
> > 1. We compare against candidates $\{\phi(0, 100), \phi(10, 100), \phi(20, 100), \ldots, \phi(90, 100)\}$
> > 2. By the periodicity property (Definition 3.1), this is equivalent to comparing $\phi(4, 10)$ against $\{\phi(0, 10), \phi(1, 10), \ldots, \phi(9, 10)\}$
> > 3. We're simply asking: "which single digit (0-9) is closest?"
> >
> > The same logic applies to every digit position—we always compare against the 10 possible digit values $\{\phi(0, 10), \ldots, \phi(9, 10)\}$, regardless of whether we're decoding the ones, tens, hundreds, or any other position.
> >
> > This design enables parallel, position-independent decoding: each digit is classified as one of $\{0, 1, \ldots, 9\}$ using the same comparison set, as illustrated in Figure 2. We have clarified this following Definition 3.6.
> >
> >
> > ---
> >
> > **Q7. How efficient is the end to end training? The manuscript claims that this method is even more efficient than the baselines since fewer tokens are used, but I imagine that computing the FoNE representation and adding it to the embedding vector creates an overhead. How long does training one epoch take in comparison to the baselines?**
> >
> > A7. Thank you for this important question about computational efficiency. The training and inference times reported in Table 1 already include the full FoNE encoding and decoding overhead.
> >
> >
> > 1. Table 1 shows FoNE achieves competitive or faster training times compared to baselines, with the encoding/decoding cost already included in these measurements. FoNE's encoding is a simple closed-form calculation (sine/cosine of pre-computed frequencies) that adds minimal overhead compared to Transformer forward/backward passes.
> >
> > 2. While FoNE adds encoding cost, it represents each number as a single token rather than multiple subword tokens. For example, "123456" requires 1 token with FoNE vs. 2 or 6 tokens with BPE or digit-wise tokenization.
> >
> > The end-to-end results demonstrate that FoNE's computational cost is practical and competitive with standard tokenization methods.

---

### Official Review · Reviewer_KuCM · 2025-11-01

**Soundness:** 3
**Presentation:** 3
**Contribution:** 3
**Rating:** 6
**Confidence:** 4

**Summary:**

The paper introduces FoNE (Fourier Number Embedding), a novel method for representing numerical values in language models. FoNE maps each number directly into an embedding using sine and cosine functions with multiple periods (powers of 10). This creates a single-token representation that encodes both the magnitude and structure of numbers.

Empirical evaluations on synthetic arithmetic tasks (addition, subtraction, multiplication) show that a small Transformer (38M parameters) trained with FoNE achieves perfect or near-perfect accuracy while being more data- and parameter-efficient than subword, digit-wise, or XVAL embeddings. The paper also discusses integration with existing models, such as continual pretraining on LLMs, and provides theoretical justifications for why FoNE preserves numeracy through modular encoding.

**Strengths:**

[S1] Clear motivation and valuable conceptual connection. I think it's a valuable connection between the previous finding that LLMs internally learn Fourier-like features and the clear need for accurate number embeddings (fragmented multi-token representations and frequency-based biases).

[S2] The empricial results are strong. Across multiple arithmetic tasks, FoNE consistently outperforms established baselines such as digit-wise, subword, and XVAL embeddings. The method achieves up to 64x higher data efficiency and strong parameter efficiency, reaching perfect or near-perfect accuracy even with a 38 M-parameter model, surpassing fine-tuned large models like Llama-3.2-1B on the same tasks. The analysis also highlights faster training and inference, fewer tokens per number, and robustness across different architectures, which together underscore the practicality and scalability of the approach.

[S3] Theoretical grounding is solid. The paper provides clear and mathematically sound explanations for why FoNE preserves numeric precision (via modular recovery and periodic embeddings).

**Weaknesses:**

[W1] Benchmarks are still synthetic. All experiments are conducted on synthetic arithmetic datasets, which may not fully reflect the complexities of real-world numeric reasoning (e.g., financial QA, table reasoning), where additional challenges might present. E.g., numbers can take different forms (e.g., "eighteen", "XVIII", $18), or with units associated (18g vs 0.018kg), and contextual grounding might be essential.

[W2] Model scale. The core experiments use very small Transformer model (38M) where text capabilities are not tested. It remains unclear whether FoNE can coordiate well with LLMs' semantic embedding space of pretrained text tokens without introducing instability. For example, there might be a misalignment between the embedidng spaces of texts and numbers; Another concern is that FoNE removed number semantics entirely: information such as "integers around 2000 is more likely to be a year number". The paper briefly argues MLP layers preserve semantics but it remains to be empirically validated. A demonstration of how FoNE interacts with pretrained embeddings or retains contextual numeric meaning would strengthen the paper’s impact.

**Questions:**

Interestingly, if one compare FoNE with the classic sinusoidal positional encoding (from Vaswani et al., 2017, Attention Is All You Need), they share very similar expressions (compared to Definition 3.2 in this paper, when switching $pos$ with the encoded number):

$$PE_{(pos, 2i)} = \sin\left(\frac{pos}{10000^{2i/d_{\text{model}}}}\right), PE_{(pos, 2i+1)} = \cos\left(\frac{pos}{10000^{2i/d_{\text{model}}}}\right)$$

In its essence, every PE is a way to encode a number (position index). Would it be possible to modify other forms of PE into number embedding methods?

---

> ### Author Response · Authors · 2025-11-22
> **Rebuttal by Authors**
>
> Thank you for the supportive feedback and thoughtful comments.
>
> **Q1.  The core experiments use a very small Transformer model. All experiments are conducted on synthetic arithmetic datasets, which may not fully reflect the complexities of real-world numeric reasoning.**
>
>
> A1. We thank the reviewer for this important question. Following previous work [1,2,3,4], we use synthetic benchmarks and small Transformer intentionally to isolate and validate the core properties of numeracy-preserving embeddings: (1) strong generalization to unseen numbers (Figure 1: 99% accuracy on 6-digit addition with only 6K examples), (2) stable representation of very large numbers (Figure 4: 60-digit addition), and (3) parameter efficient (Figure 1: small models with FoNE outperform other method on large models).  These properties cannot be reliably isolated in noisy real-world corpora where multiple confounding factors exist.
>
>
> **Real-world validation:** We train GPT-2-117M from scratch on 10B tokens from FineWeb (naturally interleaved text and numbers) with different tokenization schemes:BPE, Single-Digit, xVal, and FoNE.
>
> **Experiment Results:**
>
> Perplexity on FineWeb validation set:
>
> | Method | Validation Perplexity |
> |--------|----------------------|
> | FoNE | **46.8** |
> | xVal | 48.8 |
> | Single-Digit | 48.7 |
> | BPE | 52.6 |
>
>
>
> Perplexity of text tokens only on sequences that contain both text and numbers. ($4,984,914$ tokens in total).:
>
> | Method | Validation Perplexity |
> |--------|----------------------|
> | FoNE | **46.856** |
> | xVal | 49.160 |
> | Single-Digit | 55.922 |
> | BPE | 67.390 |
>
>
> FoNE model achieves the lowest validation perplexity on the FineWeb validation set. Furthermore, we evaluate the perplexity of text tokens only on sequences that contain both text and numbers. FoNE remains the lowest perplexity while other methods show even worse perplexity compared to their performance on all sequences.
>
> Since numbers in FineWeb appear naturally interleaved with text, this experiment shows that using FoNE throughout pretraining not only preserves the model's semantic abilities but can in fact **improve** its language modeling quality by giving better number representation.
>
>
> **Overall, our synthetic experiments establish the foundation and our real-world pretraining confirms practical applicability.**
>
>
> [1] McLeish, Sean, et al. "Transformers can do arithmetic with the right embeddings." Advances in Neural Information Processing Systems 37 (2024): 108012-108041.
>
> [2] Golkar, Siavash, et al. "xval: A continuous number encoding for large language models." arXiv preprint arXiv:2310.02989 (2023).
>
> [3] Thawani, Avijit, et al. "Representing numbers in NLP: a survey and a vision." Proceedings of the 2021 conference of the North American chapter of the association for computational linguistics: human language technologies. 2021.
>
> [4] Lee, Nayoung, et al. "Teaching arithmetic to small transformers." arXiv preprint arXiv:2307.03381 (2023).
>
> ---
>
>
> **Q2. Interestingly, if one compare FoNE with the classic sinusoidal positional encoding (from Vaswani et al., 2017, Attention Is All You Need), they share very similar expressions. In its essence, every PE is a way to encode a number (position index). Would it be possible to modify other forms of PE into number embedding methods?**
>
>
> A2. Yes. This is an excellent observation. While FoNE is motivated by empirically observed Fourier structure in learned number embeddings, for encoding, it indeed shares mathematical similarities with sinusoidal positional encoding. However, Sinusoidal PE has no decoding objective. PE is never required to reconstruct the original position index; the model only uses it to provide a smooth positional signal. In contrast, FoNE is explicitly designed so that the model can recover the exact numeric value from the embedding.
>
>
> Other PE methods such as RoPE-style relative PE only represent offsets between tokens. This is crucial for language modeling but unsuitable for numbers, where the representation must reflect the absolute scalar value, not relative distance.

---

### Official Review · Reviewer_gsTU · 2025-11-01

**Soundness:** 3
**Presentation:** 4
**Contribution:** 3
**Rating:** 6
**Confidence:** 3

**Summary:**

This paper proposes a special method called Fourier number embedding to tokenize numbers in natural language, which is defined as a list of sinusoids. A special token [NUM] is used to handle its integration with other word embeddings in both the encoding and decoding stages. Experiments show the proposed methods are effective in doing arithmetic tasks.

**Strengths:**

1. The paper identifies a well-known limitation of LLMs, their weak representations and reasoning over numbers.
2. The proposed approach is clearly explained and easy to follow, and the experiments demonstrate meaningful gains.

**Weaknesses:**

1. There are probably not sufficient discussions and experiments on the performance of tasks involving floating point and other numbers that do not necessarily represent a value (e.g., year, flight number, music pitches, etc.).
2. The contribution of the paper is quite limited on the arithmetic task, which can be done in fact by a calculator, not an LLM. A further exploration can be related to how improved number representation enhances natural language inference, in some tasks related partially to numbers.

**Questions:**

Q1&2: see weaknesses.
Q3: In the evaluation (Figure 3), is there a strict control that samples in the test set are not in the training set?
Q4: How would the author explain the result in Table 8 (authors name it Table H in the appendices), where the arithmetic ability, though higher than that of the pre-trained model, is still lower than the ones in Figure 3? How to falsify the assumption that large models that understand languages to some extent (like GPT/LLama) cannot understand numbers well, regardless of the tokenization method?

---

> ### Author Response · Authors · 2025-11-22
> **Rebuttal by Authors**
>
> We are glad that the reviewer finds our paper identifies the limitation of LLMs and experiments demonstrate meaningful gains. We hope our detailed clarification with further experiment results will clear the doubts about the significance of our work.
>
> ---
>
> **Q1. Not sufficient discussions and experiments on the performance of tasks involving floating point.**
>
> A1. We thank the reviewer for raising this point.  We want to emphasize that our paper already includes experiments on *floating-point* arithmetic—specifically, the 6-digit decimal addition task (Figure 1). FoNE achieves the same near-perfect accuracy on decimal addition as on integer addition, confirming that it handles floating-point inputs equally well.
>
> We did not repeat floating-point versions for every other arithmetic task because, in FoNE, floating-point numbers are represented identically to integers: the fractional part simply extends the Fourier series with smaller-period components.
>
> Adding fractional digits only introduces additional frequency components (e.g., $10^{-1}, 10^{-2}, \dots$) without altering the learned representation or model behavior.
>
> To validate the claim, we compared model performance on integer-format and floating-point-format versions of the same arithmetic tasks with different portions of the entire dataset. Average difference across all experiments: $0.44\\% ( \pm 0.84\\%)$. This demonstrates that FoNE encoding is representation-agnostic when digit lengths are set consistently.
>
>
> | Operation      | Data Trained | Integer | Floating-point |
> |---------------------|------------|-------------|-------------|
> | 5 digit Addition        | 1%         | 96.16       | 96.56       |
> |                     | 5%         | 99.98       | 100.00      |
> |                     | 10%        | 100.00      | 99.98       |
> | 5 digit Subtraction     | 1%         | 97.30       | 96.38       |
> |                     | 5%         | 99.98       | 99.94       |
> |                     | 10%        | 100.00      | 100.00      |
> | 3 digit Multiplication  | 1%         | 6.46        | 5.96        |
> |                     | 5%         | 38.10       | 38.30       |
> |                     | 10%        | 45.96       | 48.86       |
>
>
> ---
>
> **Q2. Not sufficient discussions and experiments on the other numbers that do not necessarily represent a value (e.g., year, flight number, music pitches, etc.).**
>
>
> A2. Thank you for raising this important point about numbers that carry semantic meaning (years, flight numbers, identifiers, etc.) rather than purely quantitative value.
>
> We train GPT-2-117M from scratch on 10B tokens from FineWeb (naturally interleaved text and numbers) with different tokenization schemes:BPE, Single-Digit, xVal, and FoNE.
>
> **Experiment Results:**
>
> Perplexity on FineWeb validation set:
>
> | Method | Validation Perplexity |
> |--------|----------------------|
> | FoNE | **46.8** |
> | xVal | 48.8 |
> | Single-Digit | 48.7 |
> | BPE | 52.6 |
>
>
>
> Perplexity of text tokens only on sequences that contain both text and numbers. ($4,984,914$ tokens in total).:
>
> | Method | Validation Perplexity |
> |--------|----------------------|
> | FoNE | **46.856** |
> | xVal | 49.160 |
> | Single-Digit | 55.922 |
> | BPE | 67.390 |
>
>
> FoNE model achieves the lowest validation perplexity on the FineWeb validation set. Furthermore, we evaluate the perplexity of text tokens only on sequences that contain both text and numbers. FoNE remains the lowest perplexity while other methods show even worse perplexity compared to their performance on all sequences.
>
> Since numbers in FineWeb appear naturally interleaved with text, this experiment shows that using FoNE throughout pretraining not only preserves the model's semantic abilities but can in fact **improve** its language modeling quality by giving better number representation.
>
>
> **Discussions:**
>
>
> This confirms that numeracy-preserving embeddings do not interfere with learning semantic numeric meanings. The model learns "2000 is often a year" and "2000 as a quantity" simultaneously—the embedding provides the magnitude, while the MLPs capture the semantic patterns.
>
> **Semantic information is learned in MLP layers, not just embeddings:** Transformer MLPs function as key-value memories that store semantic associations  [1]. While FoNE focuses the embedding on numerical magnitude, semantic context (e.g., "2000 is likely a year", "555-1234 is a phone pattern") is learned in MLP layers through exposure to contextual patterns during pretraining.
>
> [1] Geva, Mor, et al. "Transformer feed-forward layers are key-value memories." Proceedings of the 2021 Conference on Empirical Methods in Natural Language Processing. 2021.

---

> > ### Author Response · Authors · 2025-11-22
> > **Rebuttal by Authors (part 2)**
> >
> > **Q3. The contribution of the paper is quite limited on the arithmetic task, which can be done in fact by a calculator, not an LLM. A further exploration can be related to how improved number representation enhances natural language inference, in some tasks related partially to numbers.**
> >
> > A3. We appreciate this concern. While arithmetic alone can be handled by calculators, our contribution addresses a foundational capability: enabling LLMs to internally and natively represent and reason about numbers precisely [1,2,3,4]. Many real-world tasks—financial analysis, time series prediction, scientific data interpretation—require jointly understanding numerical quantities and textual context. A calculator cannot capture the semantic relationships between numbers and language (e.g., "revenue increased 23% to $4.5M" requires understanding both the percentage and absolute values in context).
> >
> > Arithmetic serves as a controlled testbed to validate that our embedding correctly encodes numeracy. We show FoNE can represent up to 60-digit numbers in a single embedding (Figure 4), providing the foundation for models to learn numeracy and semantics jointly during pretraining.
> >
> > Our experiments confirm FoNE integrates cleanly with language modeling. In continual pretraining (Table 8), MMLU remains unchanged while arithmetic accuracy significantly improves. More importantly, in our new pretraining-from-scratch experiment on 10B FineWeb tokens, FoNE achieves the lowest validation perplexity among all number tokenization schemes (BPE, Single-Digit, xVal). Since FineWeb contains naturally interleaved text and numbers, this demonstrates that numerically faithful embeddings not only coexist with semantic representations but actually **improve overall language modeling quality**.
> >
> > [1] McLeish, Sean, et al. "Transformers can do arithmetic with the right embeddings." Advances in Neural Information Processing Systems 37 (2024): 108012-108041.
> >
> > [2] Golkar, Siavash, et al. "xval: A continuous number encoding for large language models." arXiv preprint arXiv:2310.02989 (2023).
> >
> > [3] Thawani, Avijit, et al. "Representing numbers in NLP: a survey and a vision." Proceedings of the 2021 conference of the North American chapter of the association for computational linguistics: human language technologies. 2021.
> >
> > [4] Lee, Nayoung, et al. "Teaching arithmetic to small transformers." arXiv preprint arXiv:2307.03381 (2023).
> >
> >
> > ---
> >
> > **Q4. In the evaluation (Figure 3), is there a strict control that samples in the test set are not in the training set?**
> >
> >
> > A4. Yes, we strictly ensure that no samples in the test set appear in the training set. Moreover, to prevent the model from implicitly learning the commutative rule (as discussed in lines 265–269), we enforce the condition a > b when generating operands, ensuring that each equation appears in only one order and no duplicates occur between training and test sets.
> >
> >
> > ---
> >
> > **Q5: How would the author explain the result in Table 8, where the arithmetic ability, though higher than that of the pre-trained model, is still lower than the ones in Figure 3?**
> >
> >
> >
> >
> > A5: In the Table 8 experiment, we continually pretrain Llama-3.1-1B on MegaMath-Web-Pro. This dataset does not contain sufficient clean arithmetic question–answer pairs, which limits the model’s ability to learn precise computation patterns. This experiment aims to show FoNE will not hurt LLMs’ language ability while improving the arithmetic ability. However, in Figure 3 we train the model from scratch on a pure arithmetic dataset to show our method’s advantage on number representation.
> >
> >
> > Therefore, the performance in Table 8 is lower than in Figure 3 because Table 8 reflects continual pretraining on less clean arithmetic data with a simplified method, while Figure 3 uses the full FoNE method trained on a clean arithmetic dataset.
> >
> >
> > ---
> >
> >
> >
> > **Q6: How to falsify the assumption that large models that understand languages to some extent (like GPT/LLama) cannot understand numbers well, regardless of the tokenization method?**
> >
> >
> > A6. We believe that LLMs are able to understand numbers and language at the same time. In our paper, we first show that with very limited embedding and model parameters, the model is able to capture the numeracy of numbers and perform arithmetic operations. We also empirically show that with our FoNE method, the model achieves lower perplexity on text+numbers, which implies that the model is able to learn them together well when given better representations.

---

### Official Review · Reviewer_Cez7 · 2025-11-04

**Soundness:** 3
**Presentation:** 3
**Contribution:** 3
**Rating:** 6
**Confidence:** 4

**Summary:**

This paper proposes a new embedding scheme for numerical tokens, based on past empirical work showing that number embeddings often have fourier-like features. By directly encoding numbers in a pre-determined fourier-like scheme, they are able to dramatically improve an LLM's ability to do mathematical tasks.

**Strengths:**

The empirical results are super strong! They had the baselines I was looking for too (e.g., encoding each number as its own token).

**Weaknesses:**

I am curious about how this encoding scheme would impact training perplexity and stability on normal LLM training settings, where we have number prediction mixed in with regular language modeling loss. The paper talks about how this could work easily by using a special number token. I think for this work to really get adopted by a next-gen frontier or open-source model, it would require someone exploring sweeps of this at least at the small or medium pre-training scale.

I’m also interested how the authors think we should be handling cases of how messy tokenization can get in practice, e.g., real numbers look like everything from $1.000 to $1,000 to one thousand and 99/100 to 5263–36181 etc.

**Questions:**

See weaknesses!

---

> ### Author Response · Authors · 2025-11-22
> **Rebuttal by Authors**
>
> We appreciate the reviewer’s supportive comments and constructive suggestions. We have updated our manuscript to provide further details about the method and experiment setting. We hope our detailed clarification with further experiment results will clear the doubts about the significance of our work.
>
> ---
>
> **Q1. How would this encoding scheme impact training perplexity and stability on normal LLM training settings, where we have number prediction mixed in with regular language modeling loss?**
>
> A1. Thank you for this important question. We have a new pretraining experiment as you suggested to evaluate FoNE in realistic LLM training settings with mixed number and text data.
>
> We train GPT-2-117M from scratch on 10B tokens from FineWeb (naturally interleaved text and numbers) with different tokenization schemes: BPE, Single-Digit, xVal, and FoNE.
>
> **Results:**
>
> Perplexity on FineWeb validation split:
>
> | Method | Validation Perplexity |
> |--------|----------------------|
> | FoNE | **46.8** |
> | xVal | 48.8 |
> | Single-Digit | 48.7 |
> | BPE | 52.6 |
>
>
>
> Perplexity of text tokens only on sequences that contain both text and numbers. ($4,984,914$ tokens in total).:
>
> | Method | Validation Perplexity |
> |--------|----------------------|
> | FoNE | **46.856** |
> | xVal | 49.160 |
> | Single-Digit | 55.922 |
> | BPE | 67.390 |
>
>
> FoNE model achieves the lowest validation perplexity on the FineWeb validation set. Furthermore, we evaluate the perplexity of text tokens only on sequences that contain both text and numbers. FoNE remains the lowest perplexity while other methods show even worse perplexity compared to their performance on all sequences.
> Since numbers in FineWeb appear naturally interleaved with text, this experiment shows that using FoNE throughout pretraining not only preserves the model's semantic abilities but can in fact *improve* its language modeling quality by giving better number representation.
>
>
> ---
>
> **Q2. I’m also interested how the authors think we should be handling cases of how messy tokenization can get in practice**
>
> A2. We thank the reviewer for this important question. Models naturally learn to aggregate information across tokens in their early to middle layers, so our goal is to ensure the input embeddings are numerically precise—the model handles the rest.
>
> We use a standard numeric parser to detect numbers and replace each with a [NUM] token plus its parsed value. For example: "99/100 to 5263–36181" → [NUM]/[NUM] to [NUM]–[NUM] with values [99, 100, 5263, 36181]. FoNE then encodes each value and adds it to the corresponding [NUM] embedding.
>
> For "123,456", traditional tokenizers split it into ['123', ',', '456'], forcing the model to reconstruct the value from learned embeddings. Our approach yields ['[NUM]', ',', '[NUM]'] with FoNE(123) and FoNE(456)—each token already carries precise numerical structure, making mid-layer aggregation more reliable and enabling better generalization to unseen formats.

---

### Author Response · Authors · 2025-11-22
**Rebuttal by Authors**

Response to All Reviewers:

We thank reviewers [R1 (Cez7), R2 (gsTU), R3 (KuCM), R4 (pYSK)] for their thoughtful and highly supportive feedback! We were glad that the reviewers found **the problem important and timely** [R1, R2, R3], the methodology **novel and well-motivated** [R1, R3, R4], the presentation **clear and easy to follow** [R1, R2, R4], the theoretical **grounding solid** [R2, R3], and the empirical results **strong and compelling** [R1, R2, R3].

During the discussion, we are glad that **R4’s major questions have been addressed and that they have raised their score**. For their remaining concern, we add new experiments which show that simply aligning a pretrained model’s number embeddings to the FoNE representation already yields **improvements on GSM8K reasoning tasks across difficulty levels**. Please refer to our detailed response to R4 for supporting results.


We have updated the paper to incorporate constructive suggestions. We summarize the major changes based on the reviewers’ main concerns:

---

1. **Real-world LLM training** [R1, R3, R4], **semantic stability** [R1, R3], and **applicability beyond synthetic tasks** [R3, R4]

    - We add new experiments where we pretrain a Transformer model from scratch on FineWeb data using FoNE. Compared with subword, digit-wise, and XVAL embeddings, **FoNE achieves lowest perplexity** on validation data.
    - We also evaluate the perplexity of text tokens **only on sequences that contain both text and numbers**. FoNE remains the lowest perplexity while other methods show even worse perplexity compared to their performance on all sequences, which implies that bad number representation also degrades a model’s language modeling ability.
    - Importantly, semantic performance is fully preserved, indicating that FoNE aligns cleanly with the text embedding space during joint text–number pretraining and does not disrupt language understanding.
    - **We have incorporated these results into Section 5 (Q4).**


2. **Tokenization of messy numeric formats** [R1, R2, R3]

    - We clarify how FoNE handles the diverse numeric formats that appear in real text. We first apply a standard numeric-string parser that detects numbers, and replaces each detected number with a single [NUM] token together with its parsed numeric value.
    - FoNE is then applied directly to the parsed number and added to the embedding of its corresponding [NUM] token.
    - Modern LLMs can aggregate information from different tokens; with FoNE, each numeric token already carries a numerically precise representation, enabling the model to aggregate and interpret numerical information more reliably than with standard embeddings.
    - **These clarifications have been added to Section 3.4.**

---

### Meta-Review · Area_Chair_pqwd · 2026-01-06

**Summary:**

## Summary
This paper proposes Fourier Number Embedding (FoNE), an embedding scheme specialized for numbers. The key idea is to treat every number as a single token, mapped into the embedding space using Fourier features. The effectiveness of this method is tested on arithmetic tasks such as addition, subtraction, and multiplication, and the method enjoys data and parameter efficiency. The method can be combined with Abacus embedding to achieve length generalization, and longer addition problems (60-digit addition) can also be handled by chunking every five consecutive digits. The paper also tries continual pretraining of LMs with FoNE and from-scratch pretraining of LMs, demonstrating that integrating FoNE into the existing LLM training pipeline can be beneficial.

## Reviewer Concerns
Major concerns raised by reviewers can be summarized as follows.
- **Synthetic benchmarks**. All reviewers pointed out that main tasks considered in numerical evaluations were synthetic arithmetic tasks. They pointed out that the method needs to be tested on more realistic benchmarks (e.g., GSM8K) involving natural languages.
- **Compatibility with LLMs**. Reviewers KuCM and pYSK expressed concerns about whether FoNE can be incorporated well with LLMs (possibly via fine-tuning) without introducing instability.
- **Various forms of numbers**. Reviewers Cez7, gsTU and KuCM asked how the method handles various formats of numbers, such as floating point, numbers that involve semantic information (e.g., year, flight number), or those involving units.
- **Efficiency**. Reviewer pYSK expressed concerns about the efficiency of the proposed method, based on the fact that computing the FoNE representation and adding it to the embedding vector creates an overhead.

**Reviewer Concerns:**

Unfortunately, the discussion period closed before we received feedback from three out of the four reviewers. Good news for the authors was that Reviewer pYSK engaged in the discussion and chose to raise their score to 4. Based on my reading, the authors’ responses can be summarized as follows.
- **Synthetic benchmarks**.
  - The authors added from-scratch LM training experiments in the revision. They trained GPT-2-117M model from scratch with different number embeddings, and showed that FoNE has the lowest validation perplexity among four methods. They also pointed to continual pretraining experiments in the paper, which demonstrate that (simplified) FoNE can be applied to LMs to enhance arithmetic capabilities without hurting natural language abilities.
  - In the initial response, the authors did not add any GSM8K/MATH experiments because the training cost was too heavy. Reviewer pYSK was left unconvinced of the efficiency of FoNE on a large corpus of mixed text and numbers. To address the remaining concerns of Reviewer pYSK, the authors added an alternative experiment where they take a pretrained model’s number embeddings, amplify the Fourier components used in FoNE, and then fine-tune the model on downstream tasks. This scheme showed improvements on GSM8K experiments.

- **Compatibility with LLMs**. The authors clarified that FoNE inherently requires pretraining, not a drop-in finetuning method. Nevertheless, the authors emphasized that FoNE is lightweight, and compatible with standard pretraining pipelines.

- **Various forms of numbers**. The authors clarified through examples how they treat different formats of numbers using a standard numeric parser. They also clarified that the paper already handles floating point operation tasks (identical to integers under FoNE), and semantic information is captured by the MLP. For the latter part, the authors used from-scratch GPT-2 experiments to indirectly support their claim.

- **Efficiency**. The authors pointed to Table 1, which shows FoNE achieves competitive or faster training times compared to baselines, with the encoding/decoding cost already included. Reviewer pYSK responded that the efficiency claims were not fully validated on large corpora of mixed text and numbers.

**Reviewer Scores:**

The initial reviews had scores 6/6/6/2. Reviewer pYSK (initial score 2) responded to the rebuttal and raised the score to 4 after the discussion. The authors responded back with some alternative GSM8K experiments but the discussion closed before Reviewer pYSK could respond back. Here is what I expect about the reviewers’ final assessment:

- I believe that the additional from-scratch pretraining experiment and the preliminary GSM8K experiment are a great addition to the paper and the reviewers would agree with me. The additional experiments demonstrate that FoNE shows strong potential to be integrated into the LLM pipeline.
- I concur with Reviewer pYSK’s remaining concerns on the lack of full evaluation on math reasoning tasks and the method’s efficiency on such settings. It would be great if the authors could add more concrete math reasoning experiments in the next revision.
- Other than that, I think most major concerns raised by the reviewers were well-addressed in the rebuttal. This would have resulted in increases in the final score.

Overall, I believe that the paper presents an interesting approach that could be integrated into standard LM training pipelines. I recommend acceptance of this paper.

---

### Decision · Program_Chairs · 2026-01-26

Accept (Poster)